# Spatio-temporal variability and controlling factors for postglacial denudation rates in the Dora Baltea catchment (western Italian Alps)

Elena Serra[1,2], Pierre G. Valla[3,1,2], Romain Delunel[4], Natacha Gribenski[1,2], Marcus Christl[5], Naki Akçar[1]

[1]Institute of Geological Sciences, University of Bern, Bern, 3012, Switzerland
[2]Oeschger Centre for Climate Change Research, University of Bern, Bern, 3012, Switzerland.
[3]University Grenoble Alps, University Savoie Mont Blanc, CNRS, IRD, IFSTTAR, ISTerre, Grenoble, 38000, France
[4]Université Lumière Lyon 2, CNRS, UMR 5600 EVS, F-69635, France
[5]Laboratory of Ion Beam Physics, Swiss Federal Institute of Technology Zurich (ETHZ), Zurich, 8093, Switzerland

*Correspondence to*: Elena Serra (elena.serra@geo.unibe.ch)

**Abstract.** Disentangling the influence of lithology from the respective roles of climate, topography and tectonic forcing on catchment denudation is often challenging in mountainous landscapes due to the diversity of geomorphic processes in action and of spatial/temporal scales involved. The Dora Baltea catchment (western Italian Alps) is an ideal setting for such investigation, since its large drainage system, extending from the Mont Blanc Massif to the Po Plain, cuts across different major litho-tectonic units of the western Alps, whereas this region has experienced relatively homogeneous climatic conditions and glacial history throughout the Quaternary. We acquired new $^{10}$Be-derived catchment-wide denudation rates from 18 river-sand samples collected both along the main Dora Baltea river and at the outlet of its main tributaries. The inferred denudation rates vary between 0.2 and 0.9 mm/yr, consistent with previously-published values across the European Alps. Spatial variability in denudation rates was statistically compared with topographic, environmental and geologic metrics. $^{10}$Be-derived denudation rates do not correlate with modern precipitation and rock geodetic uplift. We find, rather, that catchment topography, in turn conditioned by bedrock structures and erodibility (litho-tectonic origin) and glacial overprint, is the main driver of $^{10}$Be-derived denudation patterns. We calculated the highest denudation rate for the Mont Blanc Massif, whose granitoid rocks and long-term tectonic uplift support high elevations, steep slopes and high relief and thus favour intense glacial/periglacial processes and recurring rock-fall events. Finally, our results, in agreement with modern sediment budgets, demonstrate that the high sediment input from the Mont Blanc catchment dominates the Dora Baltea sediment flux, explaining the constant low $^{10}$Be concentrations measured along the Dora Baltea course even downstream the multiple junctions with tributary catchments.

## 1 Introduction

The use of *in-situ* $^{10}$Be concentrations measured in river sediments to quantify catchment-wide denudation rates over centennial to millennial time scale is now well-established (e.g. Brown et al., 1995; Granger et al., 1996; Bierman and Steig, 1996; Granger and Schaller, 2014). $^{10}$Be concentrations are measured at the outlet of the studied basin and are inversely correlated with mean catchment denudation rate (von Blanckenburg, 2005). Widespread research has used this technique to estimate catchment denudation around the globe (see reviews in Portenga and Bierman, 2011; Willenbring et al., 2013; Codilean et al., 2018) and specifically in mountain belts such as the European Alps (Delunel et al., 2020 and references therein), with the aim of illustrating the controlling mechanisms on recent ($10^2$-$10^5$ years) denudation dynamics and assessing the respective roles of climate and tectonics forcing.

In mountainous areas, the climatic imprint on the Earth's surface denudation has been recognized over both long and short timescales. Over the Late Cenozoic to Quaternary, temperature fluctuations, increased precipitation and glaciations have participated in a worldwide increase in erosion rates (e.g. Peizhen et al., 2001; Herman et al., 2013). In the European Alps, climate forcing has led to significant topographic modification through increased slope steepness and relief (e.g. Valla et al., 2011), which in turn triggered a postglacial erosional response (Norton et al., 2010; Valla et al., 2010; Glotzbach et al., 2013;

Dixon et al., 2016). Over recent timescales ($10^2$-$10^3$ years), climate has exerted a control on denudation rates through precipitation and associated runoff (Moon et al., 2011; Bookhagen and Strecker, 2012) and by governing temperature/precipitation-dependent glacial and periglacial erosion processes (e.g. Delunel et al., 2010; Deline et al., 2014). Alternatively, other studies have shown a dominant litho-tectonic control on denudation rates (e.g. Cruz Nunes et al., 2015).

Rock-uplift and denudation rates are strongly coupled, with (1) erosional unloading driving uplift through isostatic rebound (Wittmann et al., 2007; Champagnac et al., 2009), and (2) tectonic rock uplift itself conditioning denudation rate by building new topographic gradient and increasing channel incision (e.g. Burbank et al., 1996; Montgomery and Brandon, 2002; Binnie et al., 2007; Godard et al., 2014). In transient landscapes such as recently deglaciated alpine settings, the topographic relief has not reached a steady state equilibrium between rock-uplift and denudation (e.g. Schlunegger and Hinderer, 2003; Delunel

et al., 2020) and bedrock lithology may exert a significant control on millennial catchment denudation rates through its structure and erosional resistance (erodibility, Kühni and Pfiffner, 2001). More resistant lithologies have contrasting potential controls on denudation, (1) either decreasing denudation rates because of rock-mechanical strength (Scharf et al., 2013), (2) or promoting higher denudation rates by sustaining steep topography (Norton et al., 2011).

In the European Alps, the large-scale compilation of catchment-wide denudation rates by Delunel et al. (2020) highlighted (1)

the first-order correlation between denudation rate and mean catchment slope (derived from glacial imprint on Alpine topography), (2) the absence of relationship between modern climate and denudation patterns and (3) a significant correlation between rock uplift and denudation for >100-km$^2$ catchments. This compilation also pointed at a rather weak control of bedrock lithology on denudation, with the lowest rates in the low-elevation foreland areas (with clastic sedimentary lithologies) and highest rates in the high-elevation crystalline parts (with gneissic, granitic or metamorphic lithologies) within the core of the

Alps. This trend, however, was not investigated further, since, at the scale of the European Alps, it appeared difficult to disentangle the relative influence of bedrock erodibility, topography and tectonic forcing on denudation rate as these are closely interrelated. Our study thereby aims to further explore the potential links and controls between climatically-driven topography, tectonic uplift and bedrock erodibility on the efficiency of denudation processes by investigating spatial variability of $^{10}$Be-derived denudation rates within the Dora Baltea (DB) catchment (western Italian Alps; Fig. 1). The DB catchment appears the

ideal setting for this investigation, since its large drainage system, extending from the Mont Blanc Massif (4808 m a.s.l.) to the Po Plain (around 200 m a.s.l.), cuts across the main litho-tectonic units of the western Alps (Fig. 2). Relatively similar climatic gradients and glacial history but variable bedrock lithology and geodetic uplift within the DB catchment and its tributaries allow us to assess how spatial variability in bedrock erodibility between litho-tectonic units may participate in controlling catchment topography and $^{10}$Be-derived denudation rates.

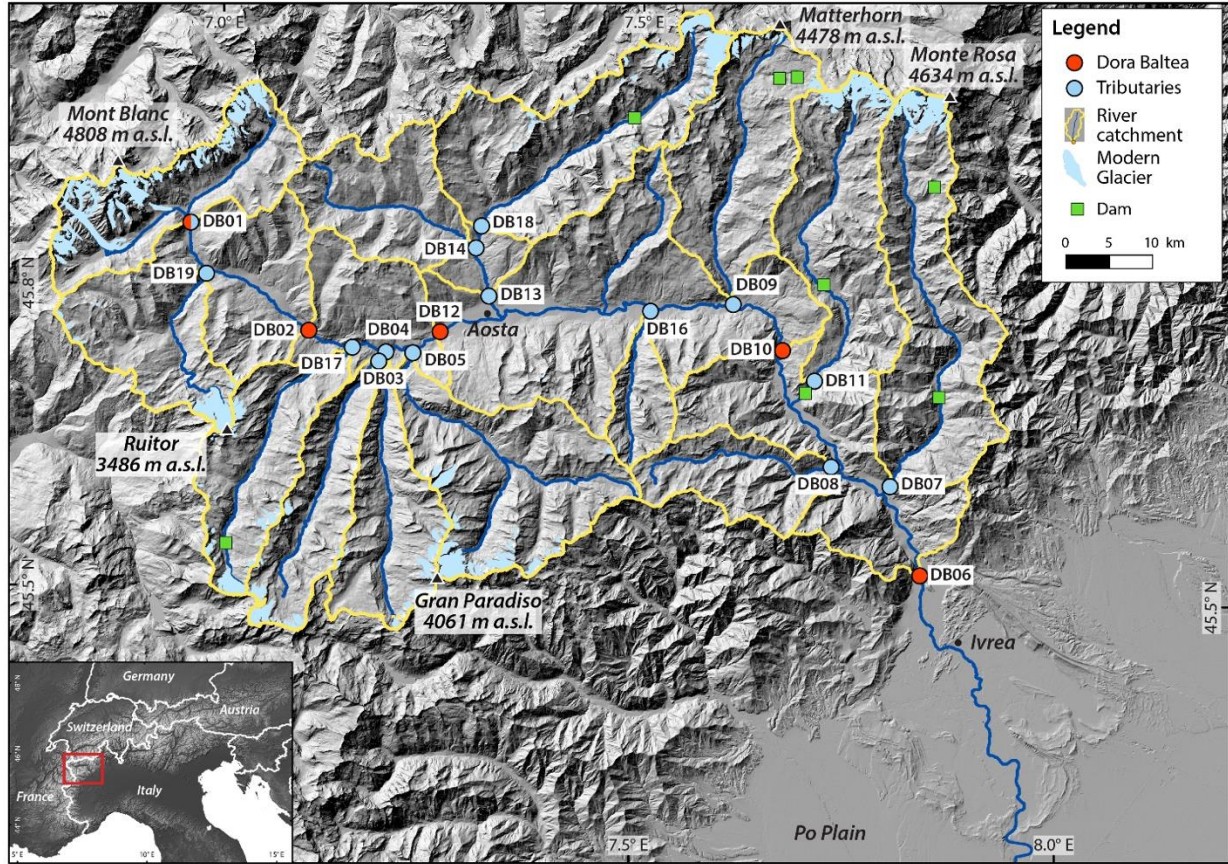


**Figure 1:** Study area with investigated Dora Baltea and main tributary river catchments (mosaic DEM from Regione Autonoma Valle d'Aosta, Regione Piemonte, swisstopo, and Institut Géographique National). Red and light blue circles indicate locations of river-sediment samples collected along the Dora Baltea river and at the outlet of the main river tributaries, respectively (for DB01 red-light blue circle as both along the Dora Baltea and considered as an individual tributary). Solid yellow lines delimit the catchments upstream of each sampling

location (sample names indicated in white box). Present-day glaciers (GlaRiskAlp Project, http://www.glariskalp.eu), main topographic peaks and dams are indicated. Inset shows location of the DB catchment (red open box) within the European Alps.

## 2 Study area

The Dora Baltea (DB) catchment is a large drainage system of ~3900 km$^2$ located in the western Italian Alps (Fig. 1). Over a 170-km long distance, the DB river flows NW-SE from the Mont Blanc Massif to the Po Plain, and drains several tributaries

connected to major >4000-m Alpine peaks (e.g. Mont Blanc, Monte Rosa, Matterhorn and Gran Paradiso; Fig. 1). Present-day mean annual temperatures range from -10°C in high-elevation zones to 15°C at valley bottoms (Regione Autonoma Valle d'Aosta, 2009). Precipitation varies between the semi-arid conditions prevailing at low elevations in the central part of the DB valley (mean annual precipitation of 400-500 mm/yr) and the wet conditions in the high-elevation internal valleys (Isotta et al., 2014). Higher mean annual precipitation values are observed in the Mont Blanc Massif (around 1800 mm/yr) compared to

the north-western and southern sectors of the DB catchment (around 1400 mm/yr for Matterhorn and Monte Rosa area, and around 1150 mm/yr in the Grand Paradiso; Isotta et al., 2014). Present-day glaciers cover 3.6% of the total DB area, and are distributed within the upstream high-elevation parts of DB tributary catchments (Fig. 1), with terminus glacier elevations ranging from 2601 to 2800 m a.s.l. (data from 2005; Diolaiuti et al., 2012).

The geology of the DB catchment is complex, since its drainage network cuts across the main litho-tectonic units of the western

Alps, recording the long-term collisional history between the European and Adriatic plates (e.g. Dal Piaz et al., 2008; Perello et al., 2008; Polino et al., 2008; Fig. 2). West of the Penninic Frontal Thrust, the European basement is exposed in the granitoid of the Mont Blanc External Massif and its Helvetic sedimentary cover. Bedrock units belonging to the thinned European crust (gneisses and schists of the Briançonnais basement and its terrigeneous to carbonate metasedimentary cover, high-pressure gneisses of the Internal Massifs), the Tethyan oceanic crust (meta-ophiolite and calcschists of the Piedmont units) and the

Adriatic margin (Austroalpine gneisses and eclogitic micaschists) are exposed roughly from NW to SE across the axial belt, delimited by the Penninic Frontal Thrust to the NW and the Insubric Fault to the SE (Fig. 2). Long-term ($10^6$-$10^7$ years) exhumation rates from bedrock apatite fission-track data are higher in the western sector of the DB catchment (0.4-0.7 km/Myr for the External zones, west of Internal Houiller Fault; Fig. 2) than in the east (0.1-0.3 km/Myr for the Internal zones, i.e. between the Internal Houiller and the Insubric Faults, Malusà et al., 2005; Fig. 2). A similar pattern has been illustrated by short-term sediment budgets inferred from detrital apatite fission-track (Resentini and Malusà, 2012) and sediment gauge (Bartolini et al., 1996; Vezzoli, 2004; Bartolini and Fontanelli, 2009) data. Modern geodetic rock uplift also appears spatially variable within the DB catchment, with rates up to 1-1.6 mm/yr in the Monte Rosa, Mont Blanc and Ruitor areas, around 0.6-0.7 mm/yr in the axial belt and in the Gran Paradiso Massif, while decreasing to 0.2 mm/yr in the Po plain (Sternai et al., 2019). For the entire DB catchment, a $^{10}$Be-derived denudation rate of 0.6 mm/yr was obtained by Wittmann et al. (2016, sample T12).

The DB catchment was repeatedly glaciated during the Quaternary, with major glaciers covering most of the catchment with the exception of the highest peaks (~3000 km², >1000 m thick; Serra et al., 2022) and extending into the Po Plain during the Last Glacial Maximum (LGM, ca. 26-19 ka; Clark et al., 2009). The tributary glaciers already retreated in their upper valley catchments during the Lateglacial climatic oscillations (14-12 ka; Baroni et al., 2021; Serra et al., 2022). As shown by the present-day topography (Fig. 1), postglacial fluvial dissection and hillslope processes have locally re-shaped the glacial landscape, with the development of V-shape valley profiles and the deposition of large alluvial fans and infills along the main valleys.

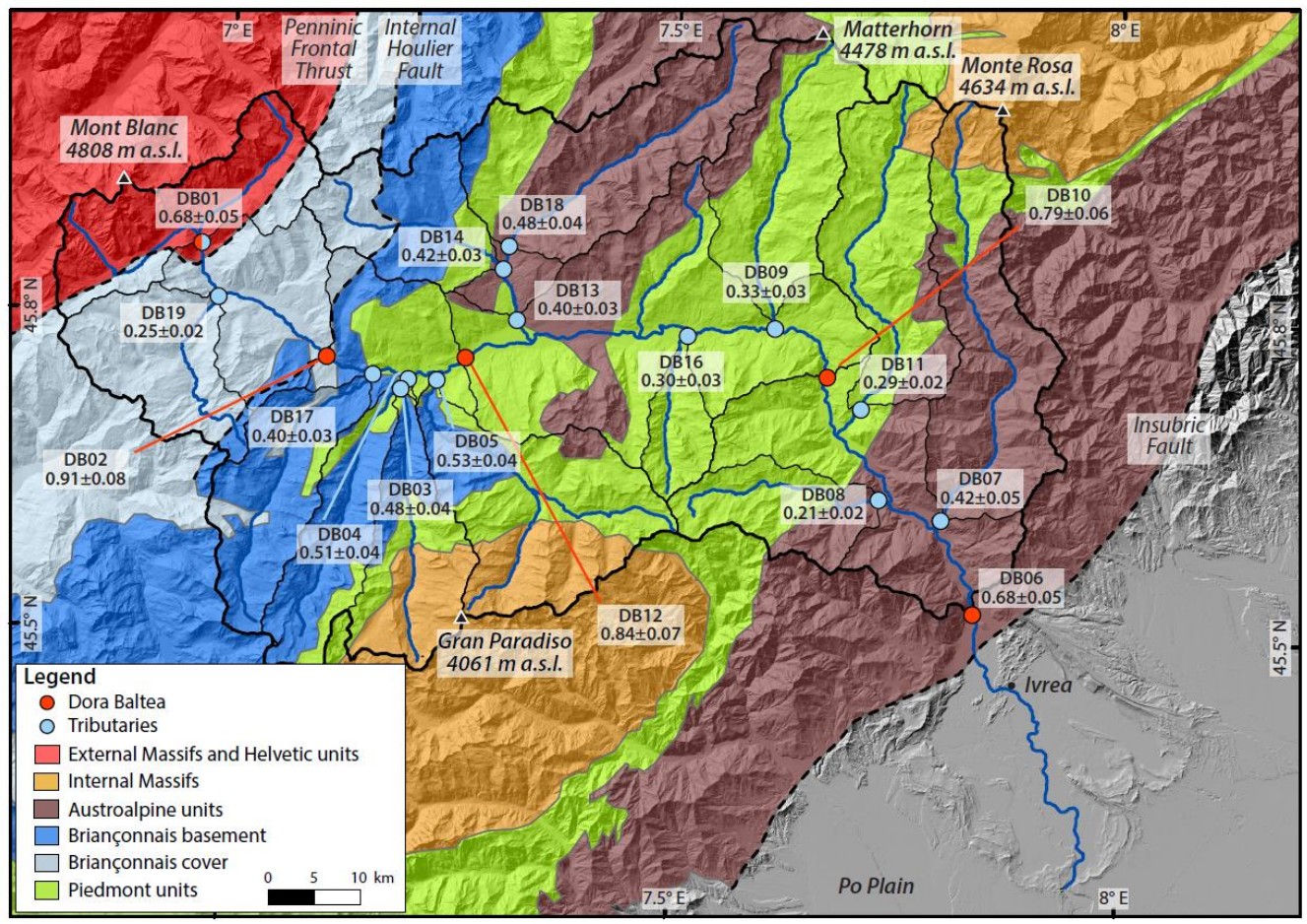

**Figure 2.** Simplified litho-tectonic map of the study area with output catchment-wide denudation rates (±1σ, mm/yr) reported at sampling locations (catchment boundaries in solid black lines). Major litho-tectonic domains and structural features (dashed lines) of the western European Alps are shown (modified map after Resentini and Malusà, 2012). Output catchment-wide denudation rates are corrected for topographic, LIA-glacier and snow shielding, and for quartz-content (see text and Table 1 for details).

## 3 Methods

### 3.1 [10]Be-derived catchment-wide denudation rates

Eighteen river-sand samples were collected within the DB catchment, 5 along the main DB river and 13 at the outlet of the main tributaries (Fig. 1). One sample (DB06) was collected at the same location as sample T12 from Wittmann et al. (2012) to assess for the possible temporal variability of the in-situ [10]Be signal exported by the DB river. Around 20-50 g of pure quartz were extracted from the 250-400 μm grainsize fraction, following sieving, magnetic separation and leaching in diluted HCl, $H_3PO_4$ and HF (detailed protocol reported in Akçar et al., 2017). The purified quartz was dissolved in concentrated HF after

addition of around 200 μg of [9]Be carrier (Table S1), and Be extraction was performed through anion and cation exchange column chemistry (Akçar et al., 2017). Measurements of [10]Be/[9]Be ratios were performed at ETH Zürich with the MILEA AMS system (Maxeiner et al., 2019), and normalized to the ETH in-house standards S2007N and S2010N (isotope ratios $28.1 \times 10^{-12}$ and $3.3 \times 10^{-12}$, respectively; Christl et al., 2013). Calculated [10]Be concentrations (Table 1) were corrected using a full process blank [10]Be/[9]Be ratio of $2.96 \pm 0.32 \times 10^{-15}$.

Spatially-averaged [10]Be production rates were derived from pixel-based calculations conducted with the Basinga 'Production rates' GIS tool (Charreau et al., 2019), using a 35-m resolution DEM from Regione Autonoma Valle d'Aosta and Regione Piemonte as input for catchment topography. For production rate calculations, we used the Lal/Stone time-dependent scaling model (Lal, 1991; Stone, 2000), which integrates corrections for atmospheric pressure and geomagnetic field fluctuations according to the ERA-40 reanalysis database (Uppala et al., 2005) and the Muscheler's VDM database (Muscheler et al.,

2005), respectively. The total [10]Be surface production rate at each DEM cell of the studied catchments was calculated based on a [10]Be production rate at sea-level and high-latitude (SLHL) of $4.18 \pm 0.26$ at $g^{-1}$ $yr^{-1}$ (Martin et al., 2017) considering relative contributions of 98.86, 0.87 and 0.27% by neutrons, slow muons, and fast muons, respectively (Charreau et al., 2019, after Braucher et al., 2011, and Martin et al., 2017). Attenuation length values of 160, 1500 and 4320 g $cm^{-2}$ were used for neutrons, slow muons, and fast muons, respectively (Charreau et al., 2019, after Braucher et al., 2011), and a rock density of 2.7 g $cm^{-3}$

was assumed. Denudation-rate uncertainties (one-sigma external) were estimated only based on values and relative errors of [10]Be concentrations and cosmogenic production rates from neutron and muons (Eq. 5 in Charreau et al., 2019). Integration times associated to the denudation rates (i.e. the time needed to erode the uppermost ~0.6 m of bedrock and ~1 m of soil; von Blanckenburg, 2005) were calculated as mean estimates (no uncertainty propagated).

Catchment-averaged production rates were corrected for (1) topographic shielding, (2) quartz-content, (3) Little Ice Age (LIA,

1250-1860 CE) glacier cover, and (4) snow shielding (Charreau et al., 2019). Catchment topographic shielding was computed with the 'toposhielding' Topotoolbox function (Schwanghart and Scherler, 2014), following the method of Dunne et al. (1999) and Codilean (2006). We acknowledge the recent publication by DiBiase (2018) suggesting no need to correct for topographic shielding when calculating catchment-wide [10]Be denudation rates. Our [10]Be production rates were however corrected for topographic shielding to follow a conservative approach similar to the recent Alpine compilation study by Delunel et al. (2020).

As reported in Table 1, mean topographic shielding values obtained within the DB catchment are all very similar (~0.95), implying that neglecting the topographic-shielding correction would result in similar output rates generally within error estimates.

Based on the 1/100,000- and 1/250,000-scale digital geological maps from Regione Autonoma Valle d'Aosta and Regione Piemonte, respectively, we mapped and excluded from the [10]Be production-rate calculation the catchment areas covered by

mafic and non-siliceous sedimentary (carbonate) bedrocks (Fig. S1), based on the assumption that they do not provide (or to a minor extent) quartz grains to the fluvial routing system. Crystalline bedrocks and Quaternary deposits (Fig. S1) were instead considered as quartz-bearing lithologies in our approach. In addition, we excluded areas with slope < 3°, assuming that they are likely not linked to the stream network or act as storage/transfer areas and therefore do not reflect catchment denudation (Fig. S1; Delunel et al., 2010). In order to estimate shielding correction due to glacier cover, [10]Be production rates were set to

null for areas covered by LIA glaciers (GlaRiskAlp Project, http://www.glariskalp.eu; Fig. S1). This conservative approach assumes sufficient ice thickness for complete cosmic-ray shielding (e.g. Delunel et al., 2010; Wittmann et al., 2007). Shielding correction factors for snow cover were calculated as function of the average elevation for each individual catchment, by applying an empirical model reported in Delunel et al. (2020) and predicting snow-shielding factors as a function of elevation for the European Alps. The obtained average snow-shielding correction factors vary between 0.82 and 0.87 and were combined to the topographic-shielding corrections as scaling factors for each sub-catchment. Catchment-wide denudation rates were then obtained using the previously-calculated catchment-averaged $^{10}$Be production rates and the measured $^{10}$Be concentrations (Table 1), with the Basinga 'Denudation rates' GIS tool (Charreau et al., 2019).

### 3.2 Topographic, environmental and geological metrics

We investigated potential drivers conditioning the observed spatial variability in DB catchment-wide denudation rates, and to this aim we performed topographic analyses, and extracted environmental and geological variables of the quartz-bearing areas (Fig. S1) for each investigated catchment through an ArcGIS-Matlab routine (Delunel et al., 2020).

Topographic analyses were conducted using a 35-m resolution DEM (Regione Autonoma Valle d'Aosta and Regione Piemonte). We calculated drainage area, mean elevation, mean slope, percentage of slopes steeper than 40°, geophysical relief, and hypsometric integral for the quartz-bearing areas of the individual catchments (Table 2). For slope analyses, the 'gradient8' Topotoolbox function was used (Schwanghart and Scherler, 2014), returning the steepest downward gradient of the 8-connected neighbouring cells of the DEM. The percentage of catchment slope steeper than 40° was calculated as indicative of the areal proportion of oversteepened threshold landscape (DiBiase et al., 2012). The geophysical relief (i.e. averaged elevation differences between a surface connecting highest topographic points and the current topography; Small and Anderson, 1998) was calculated in ArcGIS using a 5-km radius sampling window, and can be used as an indicator of past landscape change or potential for locally increased erosion (see Champagnac et al., 2014 for discussion). The hypsometric integral was computed based on Eq. 1 from Brocklehurst and Whipple (2004) and is inversely related to the stage of landscape evolution (i.e. more evolved landscapes, whose high-elevation areas have been eroded, present low hypsometric integrals).

In addition, we extracted average values of the following environmental variables for the quartz-bearing areas of each individual catchment. Average annual precipitation was obtained from the 5-km resolution grid of mean annual Alpine precipitation from Isotta et al. (2014), in order to investigate the potential influence of modern precipitation/runoff on denudation dynamics. Percentage of bare-rock area was estimated from the extent of class 30 ("bare bedrock") of the 100-m resolution CORINE Land Cover Inventory (2018), to consider if catchment areas with null to low soil/vegetation cover have higher denudation rates. LIA-glacier areal cover was calculated based on the LIA-glacier extent mapped within the GlaRiskAlp Project (http://www.glariskalp.eu), to assess the influence of modern to historical glacial and periglacial processes on output $^{10}$Be-derived denudation rates (Delunel et al., 2010). Mean LGM ice-thickness and areal percentage of each catchment above the LGM Equilibrium Line Altitude (ELA) were estimated using the LGM paleo-glacier reconstruction of the DB system (70-m resolution, LGM ELA at 2103 m a.s.l.; Serra et al., 2022). Both metrics give an indication on the LGM glacial imprint on topography and subsequent potential for postglacial erosion response (Norton et al., 2010; Salcher et al., 2014; Delunel et al., 2020). Catchment average temperature was not estimated since, at the relatively constant latitude of the investigated catchments, temperature variability directly follows catchment hypsometric distribution and thus relates to catchment elevation, which is one of the metrics analysed in our study.

Lastly, we extracted geological variables for the studied catchments. Based on the simplified litho-tectonic map of the DB catchment (Fig. 2), modified after Resentini and Malusà (2012), we estimated the relative proportion of the different litho-tectonic units within the quartz-bearing areas of each catchment. Catchment-average geodetic uplift rates were considered using the 30-km resolution interpolation grid from Sternai et al. (2019), here downscaled to 600-m resolution grid (Delunel et al., 2020).

# 4 Results

## 4.1 Spatial variability in catchment-wide denudation rates

Calculated catchment-wide denudation rates vary according to the applied production-rate correction factors (Table S2).

Uncorrected denudation rates (i.e. including only mean catchment topographic shielding and excluding areas with slope <3°) range between 0.27±0.02 and 1.49±0.13 mm/yr, while rates obtained by applying all corrections vary between 0.21±0.02 and 0.91±0.08 mm/yr (Table 1 and Fig. 2). Significant production-rate corrections were obtained when taking into account snow shielding and LIA-glacier cover (up to 17 and 42% reduction compared to uncorrected [10]Be production/denudation rates, respectively), especially for catchments with high mean elevations and associated large LIA-glacier coverage (DB01 in particular; Table 2). Lower corrections were obtained when considering quartz-bearing areas (maximum 10% reduction in output [10]Be production/denudation for catchments DB08 and 11, where relatively abundant sedimentary and mafic bedrocks occur; Fig. S1). All corrections combined together lead to reduction in [10]Be production/denudation rates of 16-53% compared to the uncorrected estimates (Table S2). Hereafter, we consider [10]Be production/denudation rates obtained by applying all corrections (Table 1 and Fig. 2), in order to maintain a conservative approach as in the recent Alpine compilation study (Delunel et al., 2020) and we acknowledge that our corrected catchment-averaged [10]Be production and denudation rates should be considered as minimum estimates.

| Sample | Location WGS 84 (dd N/ dd E) | Elevation (m a.s.l.) | [10]Be concentration (x $10^4$ at g[-1]) | Topographic shielding | Mean production rate (at g[-1] yr[-1]) | | Denudation rate (mm/yr) | | Integration time (yr) | |
|---|---|---|---|---|---|---|---|---|---|---|
| | | | | | Uncor. | Corr. | Uncor. | Corr. | Uncor. | Cor. |
| *DB01* | 45.8040/ 6.9653 | 1230 | 1.29±0.07 | 0.92 | 29.4 | 13.7 | 1.45±0.11 | 0.68±0.05 | 400 | 900 |
| *DB02* | 45.7167/ 7.1101 | 783 | 1.08±0.07 | 0.94 | 25.1 | 15.3 | 1.49±0.13 | 0.91±0.08 | 400 | 700 |
| DB03 | 45.6925/ 7.1935 | 699 | 2.35±0.14 | 0.94 | 28.6 | 18.0 | 0.76±0.06 | 0.48±0.04 | 800 | 1300 |
| DB04 | 45.7003/ 7.2019 | 664 | 2.20±0.11 | 0.94 | 27.5 | 17.8 | 0.78±0.06 | 0.51±0.04 | 800 | 1200 |
| DB05 | 45.7001/ 7.2337 | 638 | 2.05±0.10 | 0.95 | 27.8 | 17.4 | 0.85±0.06 | 0.53±0.04 | 700 | 1100 |
| *DB06* | 45.5228/ 7.8375 | 251 | 1.54±0.08 | 0.95 | 22.6 | 16.1 | 0.94±0.07 | 0.68±0.05 | 600 | 900 |
| DB07 | 45.5962/ 7.7956 | 325 | 2.25±0.26 | 0.95 | 22.2 | 15.3 | 0.61±0.07 | 0.42±0.05 | 1000 | 1400 |
| DB08 | 45.6118/ 7.7310 | 373 | 4.85±0.21 | 0.96 | 20.2 | 15.9 | 0.27±0.02 | 0.21±0.02 | 2300 | 2900 |
| DB09 | 45.7352/ 7.6124 | 465 | 3.34±0.18 | 0.96 | 24.4 | 17.5 | 0.46±0.04 | 0.33±0.03 | 1300 | 1800 |
| *DB10* | 45.7079/ 7.6713 | 375 | 1.35±0.07 | 0.95 | 23.5 | 16.5 | 1.12±0.09 | 0.79±0.06 | 500 | 800 |
| DB11 | 45.6830/ 7.7115 | 546 | 3.47±0.20 | 0.96 | 24.5 | 16.0 | 0.44±0.04 | 0.29±0.02 | 1400 | 2100 |
| *DB12* | 45.7183/ 7.2651 | 594 | 1.26±0.08 | 0.95 | 25.6 | 16.4 | 1.30±0.11 | 0.84±0.07 | 500 | 700 |
| DB13 | 45.7482/ 7.3224 | 753 | 2.79±0.12 | 0.95 | 24.4 | 17.2 | 0.56±0.04 | 0.40±0.03 | 1100 | 1500 |
| DB14 | 45.7882/ 7.3061 | 600 | 2.83±0.13 | 0.96 | 22.1 | 18.5 | 0.50±0.04 | 0.42±0.03 | 1200 | 1400 |
| DB16 | 45.7386/ 7.4292 | 524 | 3.88±0.35 | 0.94 | 23.6 | 18.7 | 0.38±0.04 | 0.30±0.03 | 1600 | 2000 |
| DB17 | 45.7039/ 7.1622 | 689 | 2.71±0.13 | 0.95 | 27.2 | 17.3 | 0.63±0.05 | 0.40±0.03 | 1000 | 1500 |

| DB18 | 45.7039/7.1622 | 689 | 2.30±0.12 | 0.94 | 27.0 | 17.2 | 0.75±0.06 | 0.48±0.04 | 800 | 1300 |
| DB19 | 45.7619/6.9873 | 1005 | 4.19±0.15 | 0.96 | 25.8 | 16.0 | 0.39±0.03 | 0.25±0.02 | 1500 | 2400 |

**Table 1:** River-sediment sample locations, measured [10]Be concentrations, calculated mean catchment [10]Be production rates, and output denudation rates and apparent ages. Sample coordinates are given in decimal degrees (dd), and sample names collected along the main DB river are written in italics. Production rate estimates (and derived denudation rates / apparent ages) are provided for (1) topographic shielding correction (column labeled "Uncor.") and (2) including corrections for topographic shielding, snow and LIA-glacier shielding and for quartz-content (column labelled "Cor."). Mean catchment [10]Be production rates (and derived catchment denudation rates) obtained by applying each individual correction are reported in Table S2.

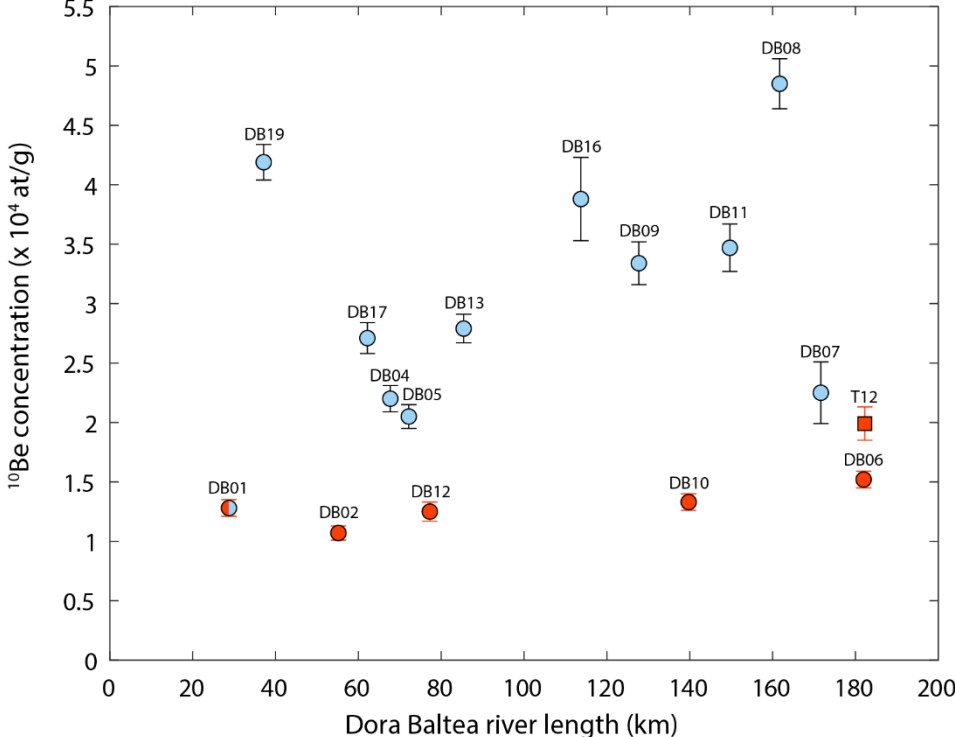

**Figure 3:** Downstream evolution of river-sand [10]Be concentrations in the Dora Baltea (DB) catchment. Data are plotted versus distance from the main DB source (upper Val Veny, right tributary upstream DB01). In red are samples collected along the main DB river, in light blue are samples at the outlet of tributaries (Fig. 1 for locations). [10]Be concentration of sample T12 (red square) from Wittmann et al. (2016) is also shown for discussion. Samples DB03, 14 and 18 are omitted since they do not directly connect to the main DR river, but note that their [10]Be concentrations are similar to those of the tributaries into which they drain (DB04 and DB13, respectively; Fig. 2 and Table 1).

[10]Be concentrations measured in the riverine samples collected along the main DB river (DB01, 02, 12, 10, 06) are the lowest and homogeneous along the DB course (1.1-1.5 x10⁴ at/g; Fig. 3). A slightly-increasing trend in [10]Be concentrations with the river distance can be observed, thereby illustrating a progressive supply of more concentrated sediments by the tributaries. The only exception to this top-down trend along the DB is observed between DB01 and DB02 where the sediments **[10]Be** concentration is diluted by ~16%. A possible explanation for such a dilution may arise from shielded materials supplied by the incision of a bedrock gorge located between samples DB01 and DB02 sampling sites (and upstream of the tributary junction with catchment DB19). Samples DB01, 02, 12, 10, 06 yield the highest and almost constant denudation rates (0.7-0.9 mm/yr), showing a slightly-decreasing trend with the river distance, excluding sample DB01 (Fig. 2). Because of its generally higher elevation and steeper topography, catchment DB01 shows the maximum production rate corrections for topographic, snow shielding and LIA-glacier cover (Table S2) and consequently yield lower output denudation rate compared to the other sampling locations downstream along the main DB river (DB02, 12, 10; Fig. 2), despite similar [10]Be concentrations (Fig. 3). This suggested overcorrection of the [10]Be production rate for the DB01 catchment is reflected by the large difference between uncorrected and corrected denudation rates (1.45 and 0.68 mm/yr, respectively, Table 1). Within tributary catchments, with

the exception of sample DB01, [10]Be concentrations are higher (2.0-4.9 x10[4] at/g; Fig. 3), and calculated denudation rates are lower, generally within 0.4-0.5 mm/yr and down to 0.2-0.3 mm/yr for some catchments (DB08, 09, 11, 16 and 19; Fig. 2).

245 To quantify at first-order the relative contribution of the Mont Blanc Massif to the [10]Be signal measured along the DB river, we followed the approach reported in Delunel et al. (2014), where relative contributions of different sediment sources can be estimated based on their respective [10]Be concentrations. Although other approaches can be used, which consider nested catchments for quantification of sub-catchments denudation rates (Mudd et al., 2016), we adopted a first-order approach in this study based only on relative contributions in [10]Be concentrations between different (sub-)catchments. We based our model

250 on sample DB02 (lowest [10]Be concentration) which provides a more conservative estimate of the contribution of the Mont Blanc Massif to the [10]Be signal measured along the DB river (i.e. the potential contributions of the tributaries are maximized). River-sediment [10]Be concentrations from tributaries and along the DB river have been first normalised to the SLHL [10]Be production rate (i.e. $4.18\pm0.26$ at g$^{-1}$ yr$^{-1}$), implying that variations in normalised [10]Be concentrations represent the variability in denudation rates only. We then estimated the respective river sediment contributions of the Mont Blanc Massif and different

255 tributaries through the mixing model of Delunel et al. (2014; $C = xA + yB$, $x + y = 1$) considering (A) the normalised [10]Be concentration for river materials exported from the Mont Blanc catchment (upstream catchment DB02), (B) the averaged normalised [10]Be concentration from the upstream tributaries contributing to each sampling points along the main DB river and (C) the normalised [10]Be concentration at the sampling points along the main DB river (DB12, 10, 06). By applying this simple model, we find that the Mont Blanc Massif contributes to 90, 87 and 77% of the river-sediment [10]Be signal measured

260 respectively at locations DB12, 10 and 06, in line with overall constant [10]Be concentrations measured along the DB course (Fig. 3).

## 4.2 Catchment metrics and denudation rates

Results of catchment topographic analyses, along with estimates of environmental and geological metrics, are reported in Table 2. As for calculated catchment-wide denudation rates, DB01 (Mont Blanc Massif) also appears as an end-member with

265 maximum values in most of the reported metrics (Table 2).

| Catchment | Total drainage area (km²) | Relative quartz-bearing area (%) | Mean elevation (m a.s.l.) | Mean slope (°) | Relative abundance of slopes > 40° (%) | Geophysical relief (5-km, m) | Hypso-metric integral | Mean annual precipitation (mm) | Relative abundance of bare-rock (%) | LIA glacier cover (%) | Mean LGM ice-thickness (m) | Basin area above LGM ELA (%) | Mean geodetic rock uplift (mm yr⁻¹) |
|---|---|---|---|---|---|---|---|---|---|---|---|---|---|
| *DB01* | 189 | 88.7 | 2600 | 32.2 | 32.7 | 1632 | 0.39 | 1762 | 34.6 | 37.3 | 474 | 77 | 1.08 |
| *DB02* | 496 | 87.3 | 2288 | 28.5 | 28.2 | 1297 | 0.38 | 1475 | 24.6 | 19.8 | 577 | 62 | 1.18 |
| DB03 | 147 | 80.3 | 2499 | 28.8 | 29.6 | 1246 | 0.54 | 1088 | 35.0 | 16.9 | 441 | 77 | 0.89 |
| DB04 | 279 | 78.0 | 2387 | 28.5 | 30.4 | 1219 | 0.51 | 1071 | 32.5 | 13.0 | 476 | 71 | 0.97 |
| DB05 | 257 | 71.9 | 2419 | 29.5 | 28.2 | 1191 | 0.52 | 1045 | 30.1 | 16.3 | 427 | 71 | 0.68 |
| *DB06* | 3321 | 73.6 | 2096 | 27.7 | 25.6 | 1214 | 0.40 | 1147 | 19.4 | 10.3 | 526 | 53 | 0.95 |
| DB07 | 278 | 85.7 | 2030 | 30.2 | 26.9 | 1076 | 0.41 | 1352 | 11.1 | 8.7 | 304 | 43 | 0.67 |
| DB08 | 108 | 56.4 | 1827 | 29.3 | 29.3 | 978 | 0.52 | 1164 | 4.1 | 0.3 | 282 | 36 | 0.48 |
| DB09 | 207 | 62.0 | 2228 | 24.4 | 18.6 | 1232 | 0.45 | 1092 | 21.1 | 10.8 | 317 | 61 | 1.32 |
| *DB10* | 2464 | 77.9 | 2156 | 28.1 | 26.0 | 1234 | 0.40 | 1130 | 22.3 | 11.3 | 553 | 57 | 1.03 |
| DB11 | 226 | 48.5 | 2060 | 25.5 | 18.2 | 1211 | 0.40 | 1051 | 13.3 | 7.8 | 377 | 47 | 0.96 |
| *DB12* | 1310 | 80.7 | 2284 | 28.2 | 26.9 | 1226 | 0.40 | 1228 | 25.2 | 15.3 | 534 | 64 | 1.03 |
| DB13 | 450 | 90.6 | 2224 | 29.0 | 28.0 | 1148 | 0.46 | 1144 | 26.4 | 10.2 | 558 | 61 | 1.10 |
| DB14 | 141 | 92.4 | 2090 | 26.8 | 20.0 | 1092 | 0.45 | 1269 | 12.4 | 0.8 | 568 | 51 | 0.93 |
| DB16 | 54 | 53.4 | 2154 | 27.8 | 25.2 | 1146 | 0.62 | 949 | 18.7 | 2.1 | 369 | 58 | 0.68 |
| DB17 | 158 | 79.4 | 2335 | 27.6 | 23.7 | 1107 | 0.55 | 1047 | 26.3 | 13.9 | 467 | 70 | 1.24 |
| DB18 | 277 | 89.4 | 2406 | 30.9 | 34.0 | 1137 | 0.48 | 1123 | 36.9 | 16.4 | 493 | 72 | 1.22 |
| DB19 | 148 | 80.3 | 2313 | 24.5 | 21.1 | 914 | 0.53 | 1251 | 23.8 | 18.0 | 506 | 69 | 1.39 |

**Table 2:** Topographic, environmental and geological metrics extracted for the studied catchments (upstream of sampling river locations for [10]Be analysis). Names of sample collected along the main DB river are given in italics. With the exception of the total drainage area, all the metrics were calculated using the quartz-bearing area of each catchment. See text for details.

We compared the [10]Be-derived catchment denudation rates against topographic, environmental and geological metrics and evaluated the statistical significance of linear correlations (p-value and $r^2$; Figs. 4 and 5, Table S3). Samples along the main DB river (downstream of DB01, i.e. DB02, 12, 10, 06; Fig. 2) were excluded from the investigated correlations since their apparent denudation rates are potentially affected by cumulative drainage and sediment mixing along the DB course. Correlations were calculated both including and excluding sample DB01. Cook's distance values were also calculated in order to assess whether DB01 strongly influences the derived correlations as a potential outlier. We selected a threshold value of 3 times for DB01 Cook's distance compared to the data mean Cook's distance. As a consequence, DB01 appears as an outlier in different investigated correlations between catchment denudation rates and topographic, environmental or geological metrics (Table S3). Nevertheless, significant linear correlations (i.e. p-value <0.05) both with and without DB01 were obtained between catchment denudation rates and mean elevation (Fig. 4A), 5-km geophysical relief (Fig. 4C), the relative abundance of bare bedrock (Fig. 5B), and the percentage of area above the LGM ELA (Fig. 5D). Significant linear correlations between catchment denudation rates and mean slopes (Fig. 4B), proportions of oversteepened slopes (Table S3) and relative area covered by LIA glaciers (Fig. 5C) were instead only found when including DB01. Non-significant correlations (p-value ≥0.05) were observed between catchment denudation rates and drainage areas (Table S3), hypsometric integrals (Table S3), mean annual precipitation values (Fig. 5A), mean geodetic uplift rates (Fig. 4D) and mean LGM ice-thickness (Table S3).

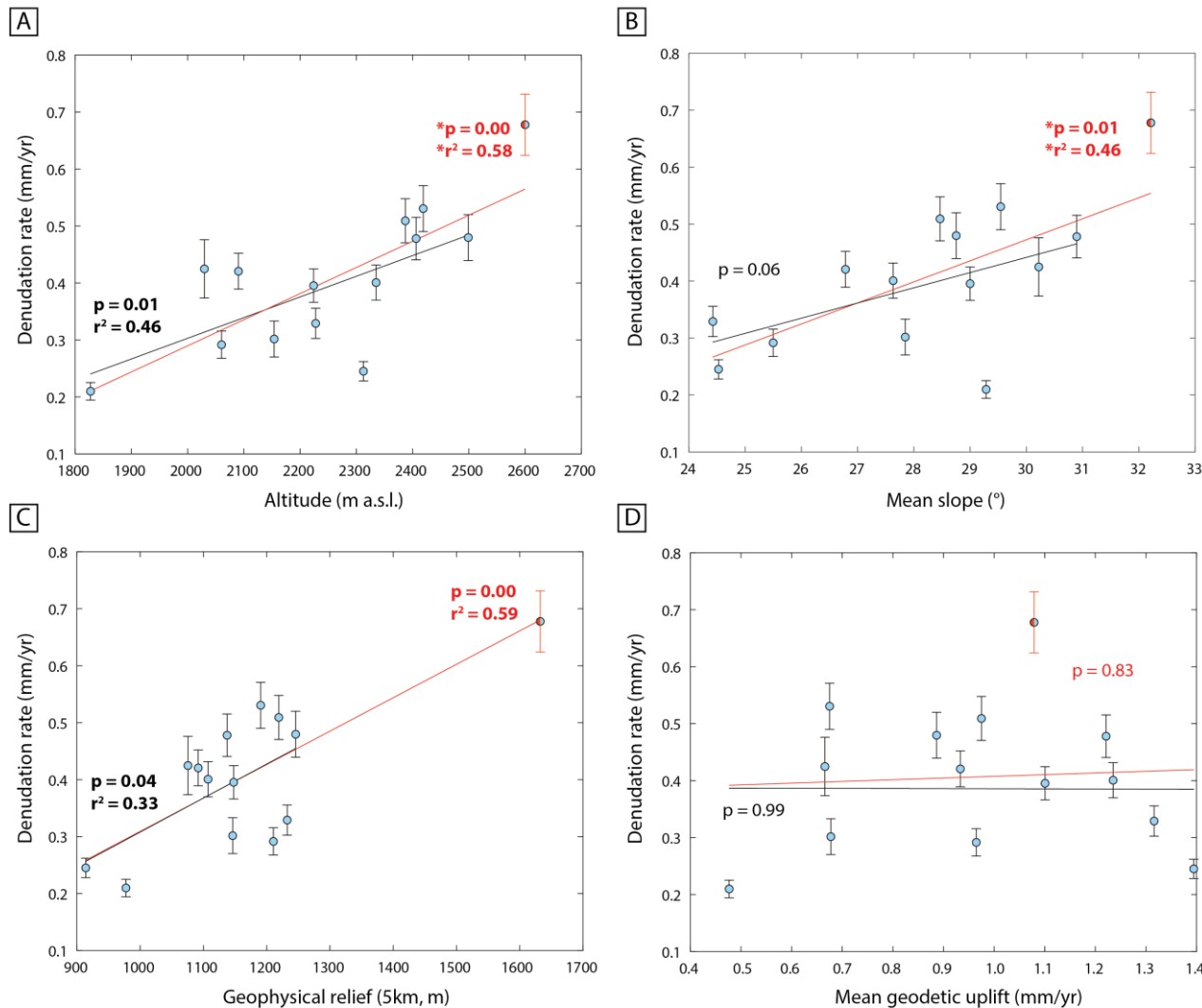

**Figure 4:** Relationships between tributary-catchment denudation rates and mean catchment (A) elevation, (B) slope, (C) 5-km geophysical relief, and (D) geodetic uplift. Linear correlations have been calculated including or not sample DB01 (red and black lines, respectively; see main text for discussion). Correlation coefficients (p-value and $r^2$) are reported for each linear regression with statistical significance (p-value < 0.05). $r^2$ is not reported for non-significant correlations (p-value > 0.05). Asterisks indicate linear correlations for which DB01 has been considered as an outlier (based on Cook's distance, Table S3).

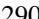

**Figure 5:** Relationships between tributary-catchment denudation rates and catchment (A) mean annual precipitation, (B) relative bare-bedrock area, (C) relative area covered by LIA glaciers, and (D) relative area above LGM ELA (2103 m a.s.l.). Correlations have been calculated including or not sample DB01 (red and black lines, respectively; see text for discussion). Correlation coefficients (p-value and $r^2$) are reported for each linear regression with statistical significance (p-value < 0.05). $r^2$ is not reported for non-significant correlations (p-value > 0.05). Asterisks indicate linear correlations for which DB01 has been considered as an outlier (based on Cook's distance, Table S3).

## 4.3 Litho-tectonic units and denudation rates

In addition to catchment metrics, we explored the potential influence of bedrock properties on the postglacial evolution of the DB catchment by analysing the correlation between tributary-catchment denudation rates and the spatial distribution of litho-tectonic units (Figs. 2 and 6; only quartz-bearing areas considered, Fig. S1). The highest denudation rates are observed for tributaries with widespread bedrock exposure of granites of the Mont Blanc External Massif and its Helvetic terrigeneous to carbonate sedimentary cover (85%; DB01: 0.68±0.05 mm/yr). Moderate denudation rates around 0.4-0.5 mm/yr are observed for catchments with abundant gneisses of the Gran Paradiso Internal Massif (41-73%; DB03, 04, 05: average denudation rate of 0.51±0.02 mm/yr), with dominant Austroalpine gneisses and eclogitic micaschists (58-85%; DB07, 13, 18: average

denudation rate of 0.43±0.03 mm/yr, with the exception of DB08 at 0.21±0.02 mm/yr) or with abundant gneisses and schists of the Briançonnais basement (63-89%; DB14, 17: average denudation rate of 0.41±0.01 mm/yr). The lowest denudation rates were obtained for tributaries dominated by meta-ophiolites and calcschists of the Piedmont units (73-100%; DB09, 11, 16: average denudation rate of 0.31±0.02 mm/yr), and by the terrigeneous to carbonate Briançonnais metasedimentary cover (87%; DB19: 0.25±0.02 mm/yr). It is however to be noted that for these tributaries (DB09, 11 and 16), the quartz-bearing areas considered in our calculations and analysis mainly consist of Quaternary deposits rather than exposed bedrocks (Fig. 2 and S1).

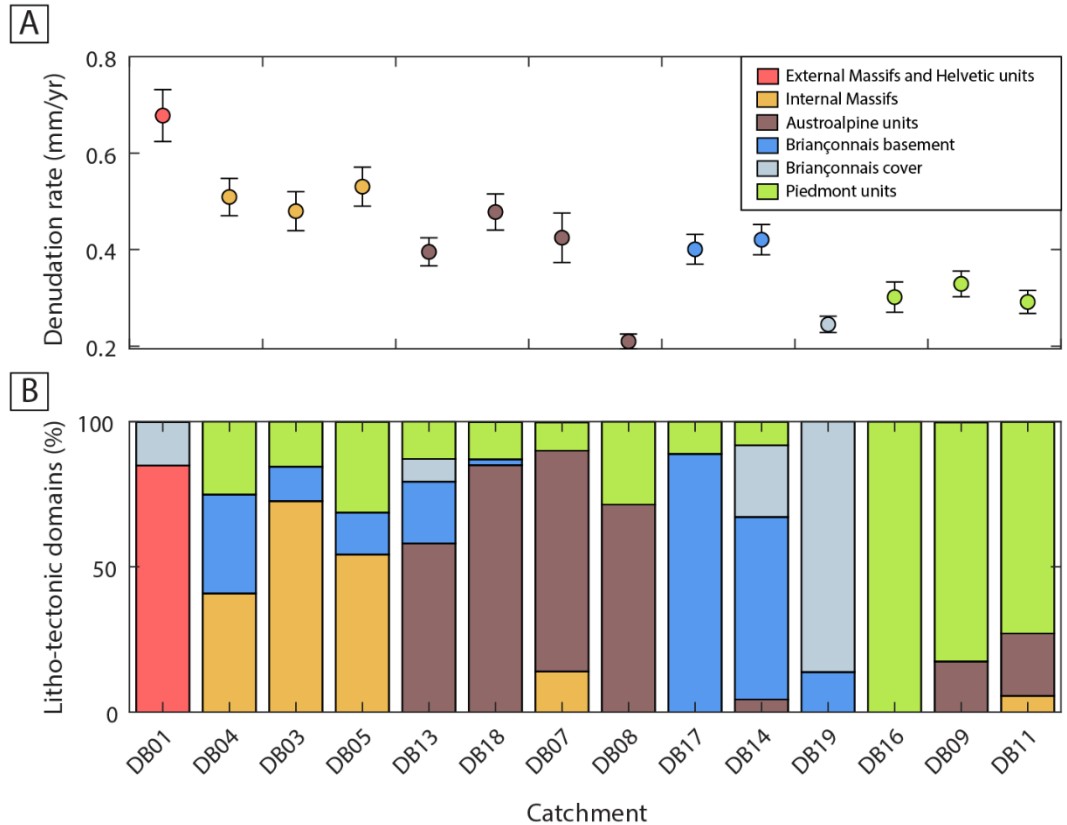

**Figure 6:** Tributary-catchment denudation rates (A) and relative proportion of litho-tectonic units within the quartz-bearing areas of the individual catchments (B). Colour code in (A) refers to the most abundant litho-tectonic unit in each individual catchment (see Figure 2 for spatial distribution of the different litho-tectonic units).

Finally, we assessed the potential influence of bedrock lithological properties on catchment morphometry, by evaluating the distribution of elevation, slope and 5-km geophysical relief for the quartz-bearing areas of each individual litho-tectonic unit (Fig. 7). Higher elevations are observed for the External and Internal Massifs (median of 2500-2700 m a.s.l.) compared to the other litho-tectonic units (median of 1900-2200 m a.s.l.; Fig. 7A). The slope distributions appear slightly higher for the External Massif, the Austroalpine units and the Briançonnais basement (median of 31-32°) than for the other units (median of 23-26°; Fig. 7B). The External Massif present the highest geophysical reliefs (median of 1700 m), while all the other litho-tectonic units have similar geophysical relief with medians varying between 1000 and 1300 m (Fig. 7C). We also evaluated the distribution of the mean geodetic uplift rates in the different litho-tectonic domains (Fig. 7D). The highest uplift rates are observed for the Briançonnais basement and cover and for the External Massif (median of 1.0-1.2 mm/yr), while the other litho-tectonic units have uplift rates with median of 0.8-0.9 mm/yr.

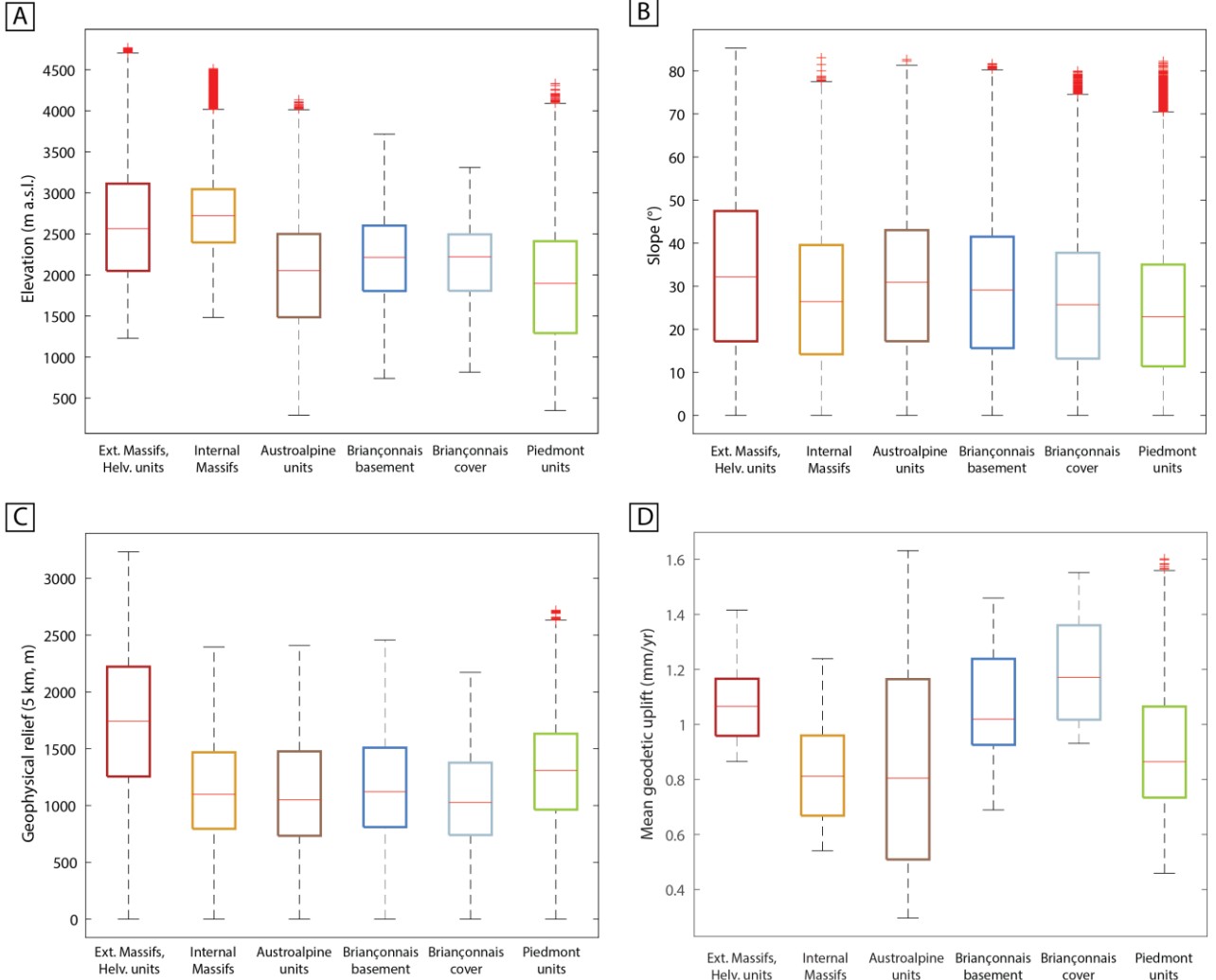

**Figure 7:** Box-and-whisker plots for the spatial distribution of elevation (A), slope (B), 5-km geophysical relief (C) and mean geodetic uplift (D) within the entire DB catchment, classified by individual litho-tectonic unit. Red horizontal line represents the median of each distribution, bottom and top of each box are the 25th and 75th percentiles. Whiskers extend up to 1.5 times the interquartile range, outliers (red crosses) are observations beyond the whiskers.

## 5. Discussion

### 5.1 Correction factors for catchment-wide $^{10}$Be production and denudation rates

As reported in Table S2, the different correction factors for quartz-content, LIA-glacier cover and snow-shielding lead to 16-53% decrease in catchment $^{10}$Be production and inferred denudation rates compared to uncorrected estimates (i.e. including only catchment-averaged topographic shielding and excluding areas with slope <3°). Such correction factors build on several assumptions and have different implications for our catchment-wide denudation rate results that are discussed hereafter. However, we should also note that the investigated correlations between denudation rates and topographic, environmental and geological metrics (Fig. 4-5) remain similar when using non-corrected denudation rates.

First, assuming Quaternary deposits as quartz-bearing lithologies is a first-order approximation, since deposits derived from mafic and carbonate-sedimentary bedrocks would bring no or minor quartz to the sediment routing system. For some tributaries dominated by these lithologies (DB09, 11 and 16, as reported in section 4.3), we nevertheless considered that Quaternary deposits may bear quartz given than the upper part of their catchments drain crystalline bedrock (Fig. S1). However, distinguishing deposit provenance/lithology in this Alpine environment, with complex glacial/periglacial systems, would require detailed field investigation and mapping, which is beyond the scope of this work. Moreover, our calculations show that

correction for quartz-bearing area has only a minimal effect on catchment-averaged [10]Be production and denudation rates, with

only up to 10% difference between uncorrected and corrected results thus overlapping within uncertainties (Table S2). Second, correction factors for LIA-glacier cover and snow shielding lead instead to significant decrease in catchment-averaged [10]Be production and thus denudation rates (up to 42 and 17%, respectively). Since sediments in sub-/proglacial environments can derive from periglacial erosion from bedrock walls/peaks and/or re-mobilization of previously exposed material (with non-zero [10]Be concentration, e.g. moraine deposits; Wittmann et al., 2007; Delunel et al., 2014; Guillon et al., 2015), assuming null

[10]Be concentration input from areas covered by LIA glaciers might lead to overcorrections of our denudation rate estimates. Uncertainties are related also to the snow-shielding correction approach. The snow-shielding vs. elevation model reported by Delunel et al. (2020) has been calibrated on snow-water equivalent records of the Swiss and French Alps, which are wetter regions compare to the DB catchment (Isotta et al., 2014). Therefore, LIA-glacier cover and snow-shielding corrections may be overestimated for the DB catchments, especially for high-elevation tributaries. In particular, as stated in section 4.1,

catchment DB01 shows the maximum corrections for both LIA-glacier cover and snow shielding (42 and 17% respectively; Table S2) and consequently relatively low output denudation rate compared to estimates obtained for catchments downstream along the main DB river (DB02, 12, 10; Fig. 2), despite similar [10]Be concentrations (Fig. 3). We therefore acknowledge that our corrected catchment-averaged [10]Be production and denudation rates for DB01 (Table 1 and Fig. 2) should be considered as minimum estimates, given the correction factors for LIA-glacier cover and snow shielding, in line with the recent

compilation over the entire European Alps (Delunel et al., 2020).

Finally, we need to estimate the impact of LGM glacial erosion on our [10]Be-derived denudation rates (Glotzbach et al., 2014; Dixon et al., 2016), since our study area has been largely glaciated during the LGM (Serra et al., 2022). Deep glacial erosion may have largely to completely zeroed [10]Be concentration on bedrock surfaces, with non steady-state [10]Be concentration depth profiles during postglacial surface exposure leading to apparent overestimate in denudation rates from [10]Be concentrations in

river sands (Glotzbach et al., 2014). However, given the deglaciation history of the DB catchment (i.e. largely deglaciated by 14-12 ka; Baroni et al., 2021; Serra et al., 2022) and the range of our [10]Be-derived denudation rates (0.2-0.9 mm/yr, Table 1 and Fig. 2), we can assess an overestimate of our [10]Be-derived denudation rates by maximum 10-15%, similar to the proposed estimate of Dixon et al. (2016) in the Eastern Alps, with a maximum 9% overestimate for slower [10]Be-derived denudation rates (~0.2 mm/yr). We thus are confident in the validity of our [10]Be-derived denudation rates (Table 1), and can exclude any

potential strong bias influencing the spatial pattern (Fig. 2) and interpretation with regards to topographic, environmental and geological metrics (Figs. 4-5).

## 5.2 Propagation of [10]Be signal along the DB course

Our results highlight the strong [10]Be-dominance of the Mont Blanc Massif (represented by sample DB02, see section 4.1 for

discussion) on downstream sediment samples collected along the DB course, below the tributary junctions (Fig. 3, Table 1). The relatively constant low [10]Be concentrations measured for samples DB01, 02, 12, 10, 06 (1.1-1.5 x10^4 at/g, Fig. 3) compared to the tributaries (2.0-4.9 x10^4 at/g), and the outcomes of our mixing model indicate unequal sediment contribution (non-balanced sediment budget; Savi et al., 2014) between the main DB stream and its tributaries. The Mont Blanc Massif appears to govern the sediment yield along the main DB river, contributing to >77% of the river-sediment [10]Be signal carried all along

the DB river.

A key factor governing the mixing and flux balance of [10]Be concentrations between river streams is the quartz flux from each stream, which is in turn influenced by (1) catchment denudation rate, (2) drainage area, (3) catchment quartz content (Carretier et al., 2015), (4) sediment storage (e.g. dams, lakes, floodplains reducing mass flux but not changing the [10]Be concentration; Wittmann et al., 2016). Our results show significantly higher denudation rate for catchment DB01 compared to other DB

tributaries (Fig. 2, Table 1). While the Mont Blanc Massif (upstream DB02 catchment) represents only a minor fraction of the

total DB catchment area (~18%), its quartz-bearing surface area appear 5-90% larger than for other tributaries (Table 2). Likewise, the sediment-provenance studies of Vezzoli et al. (2004) and the sediment-yield estimates of Vezzoli (2004) highlighted that river sands from the Mont Blanc catchment (analogous catchment to DB02) have up to ~20% higher quartz content compared to some other DB tributaries (analogous to DB09, 11, 16; Table S4) and contribute to ~62% of the quartz flux of the entire DB catchment (analogous to DB06; Table S4). Since the occurrence of dams is limited to few catchments (Fig. 1), the high quartz flux and [10]Be-signal dominance of the Mont Blanc Massif along the DB course could derive from (1) its high denudation rate (Fig. 2 and Table 1), (2) its large quartz-bearing drainage area and (3) the high quartz content of the Mont Blanc granitoid (Vezzoli, 2004). Between this three potential causes, we propose that the [10]Be-signal dominance of the Mont Blanc Massif along the DB course is mainly driven by its high denudation rather than quartz fertility or area coverage, as illustrated by the similar trend of modern denudation rates derived from sediment gauging (Hinderer et al., 2013; see also discussion in section 5.4). The high rock-slope instability and glaciogenic sediment production in the Mont Blanc Massif supply abundant low [10]Be concentration quartz to the river system, being therefore efficient in diluting the [10]Be concentration in the downstream course of the DB river. Controlling factors explaining the high denudation rate of the Mont Blanc Massif are further discussed below (section 5.3).

For the entire DB catchment, we can note that the [10]Be concentration is ~30% higher for sample T12 ($1.99\pm0.14 \times 10^4$ at/g; Wittmann et al., 2016) compared to DB06 ($1.52\pm0.08 \times 10^4$ at/g; Fig. 3), both collected at the same location (DB catchment outlet, Fig. 1) but at different time periods. The observed difference is probably related to a stochastic change in sediment sources (Lupker et al., 2012), with potentially a temporary dominant sediment input from a DB tributary catchment with higher [10]Be concentration (e.g. DB07, close location and similar [10]Be concentration as T12; Fig. 3) than from the Mont Blanc catchment. By comparing our results to the Po catchment (Wittmann et al., 2016), which drains several main river systems from the south-western Alps in addition to the DB river basin, it emerges that the low [10]Be concentration signal deriving from the Mont Blanc Massif, only slightly increasing along the DB course (from 1.1 to $1.5 \times 10^4$ at/g, Fig. 3), increases significantly soon after the DB flows into the Po river. The high [10]Be concentrations measured by Wittmann et al. (2016) in Po river-sediment samples, immediately downstream the DB confluence (samples P1 and P3: around $3.6 \times 10^4$ at/g), show that the Po river is dominated in its initial lowland flow by the high [10]Be concentration inputs from other south-western Alpine catchments (Wittmann et al., 2016).

**5.3 Controlling factors and processes on [10]Be-derived catchment denudation rates**

Our [10]Be-derived denudation rates, varying between 0.2 and 0.9 mm/yr, fit within the values obtained over the European Alps, where 95% of the considered catchments yield denudation rate values <1.2 mm/yr and rates for the Western European Alps range between 0.1 and 1.2 mm/yr (Delunel et al., 2020). Correlations with topographic, environmental and geologic metrics allowed us to identify and discuss potential controlling mechanisms for denudation-rate variability within the DB catchment, that we present here in comparison with studies conducted in other Alpine sectors.

While precipitation and rock uplift have been recognized as main drivers for Alpine denudation rates, especially for the Central Alps (e.g. Chittenden et al., 2014; Wittmann et al., 2007, respectively), their respective influence on denudation-rate variability within the DB catchment is not significant (Figs. 5A and 4D). Interestingly, it can be observed that catchment geodetic rock uplift rate is higher (20-80%) than [10]Be denudation in all the investigated tributary catchments, suggesting a net surface uplift of the DB area for recent timescales, in line with other observations across the European Alps (Norton et al., 2011; Delunel et al., 2020).

Catchment topography appears instead to have a major role in controlling the observed spatial variability in DB denudation rates. First, denudation rates are positively correlated with catchment-averaged elevation (Fig. 4A). Elevation influences denudation rates through periglacial (i.e. frost-cracking; Delunel et al., 2010) and glacial erosive processes, both increasing

with altitude due to their temperature dependency, as well as by modifying soil and vegetation cover, with bare-rock exposure being positively correlated with denudation rates (Fig. 5B). Second, correlations with slope and geophysical relief need to be considered (Figs. 4B and C). It has been previously proposed that topographic slope and geophysical relief are positively correlated to catchment denudation until a threshold slope angle of 25-30° (Montgomery and Brandon, 2002; Champagnac et al., 2014; Delunel et al., 2020). Below this threshold, denudation was shown to respond to a slope-dependent equilibrium between regolith cover production through weathering and its downslope diffusion. In oversteepened catchments, denudation rates are instead controlled by mass wasting processes (i.e. rockfalls, debris flows, landslides) which stochastically influence river-sediment $^{10}$Be concentrations. While the potential effect of slope alone is here challenging to evaluate as all the tributaries exhibit similar averaged slope values, between ~25-30° (with the exception of DB01 with average slope of ~32°, Fig. 4B), denudation rate exhibits a clear correlation with geophysical relief (Fig. 4C), which is function of both slope and elevation difference (Small and Anderson, 1998; Champagnac et al., 2014). We suggest that slope differences between the investigated catchments, while not significant, are nevertheless still close to threshold values (Fig. 4B; Delunel et al., 2020), which, when combined with elevation differences between catchments, would explain the significant relationship observed between geophysical relief and denudation rates (Fig. 4C).

Our results also show a correlation between catchment denudation rates and bedrock litho-tectonic classification (Fig. 6), which has been proposed to drive denudation through rock mechanical strength (erodibility; Kühni and Pfiffner, 2001). Similar to what has been suggested based on DB modern sediment provenance (Vezzoli et al., 2004), we observe a general trend with the highest denudation rates in catchments dominated by apparent "low erodibility" bedrocks (granite and gneiss), and the lowest rates in catchments with apparent "high erodibility" bedrocks (sedimentary and terrigeneous rocks; erodibility classes according to Kühni and Pfiffner, 2001). This trend has already been observed locally in the Eastern and Southern Alps (Norton et al., 2011) as well as at the scale of the entire European Alps (Delunel et al., 2020). Such observations were interpreted to be related to the influence of bedrock resistance on catchments morphometry (in turn connected to denudation dynamics), with the most resistant lithologies located at highest elevations and sustaining the steepest slopes/highest reliefs (Kühni and Pfiffner, 2001; Stutenbecker et al., 2016). Our results are in line with this interpretation, with the "low-erodibility" granite of the Mont Blanc External Massif supporting the highest elevation and reliefs and slightly steeper slopes (Fig. 7), where efficient geomorphic processes promote the highest catchment denudation rate (Fig. 6). On the other hand, the "high erodibility" rocks of the Briançonnais cover and of the Piedmont units present low elevation, relief and slope values, and are associated with low denudation rates. High elevation sustained by gneisses and granite of the Internal Massifs (2700 m a.s.l.; Fig. 7A) and slightly steeper slopes supported by gneisses and micaschists of the Austroalpine units and of the Briançonnais basement (30-31°; Fig. 7B) would also drive the moderate denudation rates observed in these three litho-tectonic domains (Fig. 6). Moreover, the different long-term tectonic histories between the litho-tectonic domains could also explain some of the observed variability in catchment denudation between areas west and east of the Penninic Frontal Thrust (Fig. 2). Bedrock tectonic fracturing (Molnar et al. 2007) may influence subsequent erodibility and denudation, facilitated by the exhumation of more fractured bedrock units such as the crystalline units of the Mont Blanc External Massif and its Helvetic sedimentary cover (no deep Eocene subduction during Alpine orogeny), compared to deeply-exhumed rocks of the Internal Massifs and Piedmont units (Schmid et al., 2004). No correlation appears instead between the mean geodetic uplift and the denudation rates of the different litho-tectonic units, with the highest uplift rates observed both for the fast eroding External Massif and the slow eroding Briançonnais units (Fig. 7D). This further excludes the role of geodetic uplift in controlling denudation rate variability within the DB catchment (Fig. 4D), while further supporting the dominant influence of bedrock resistance. However, the long-term (late-Miocene) high uplift rates in the Mont Blanc Massif compared to the rest of the DB catchment (Malusà et al., 2005) could be one of the causes of the high denudation rate of the Mont Blanc Massif, by sustaining high-elevations and in turn promoting efficient geomorphic processes.

Lastly, we consider the potential connection between landscape glacial imprint and catchment denudation rates. Statistically significant correlation between catchment denudation rates and catchment area proportion above LGM ELA (Fig. 5D) suggests an impact of LGM/large Quaternary glaciations on [10]Be-derived denudation rates. Most of the catchments have >50% of their area above the LGM-ELA, indicating that large glaciers persisted during the LGM (and potentially older Quaternary glacial stages), with a significant impact on catchment topography characterized by steep slopes and high reliefs (Fig. 4A-C, Pedersen

and Egholm, 2013). We hence tentatively interpret the significant correlation between denudation rates and high elevation / pronounced geophysical relief (Figs. 4B and C) to be indicative of a long-term glacial topographic control on the postglacial erosional response, as suggested by previous studies (Norton et al., 2010a; Glotzbach et al., 2013; Dixon et al., 2016). The glacial pre-conditioning of the topography has been also enhanced during postglacial times with coupled fluvial incision and hillslope processes increasing Alpine valley slopes and reliefs locally (Korup and Schlunegger, 2007; Valla et al., 2010; van

den Berg et al., 2012). Over a shorter term, the positive correlation between catchment denudation rates and LIA glacial cover (only when including DB01, Fig. 5C) suggests also an important role of Holocene to modern glacial processes in influencing catchment denudation, by contributing to high-sediment delivery (Stutenbecker et al., 2018).

By considering the above-mentioned controlling mechanisms for catchment denudation, we propose the following interpretation for the high denudation rate obtained for catchment DB01 compared to other DB tributaries (Figs. 2 and 4-5).

Catchment DB01 has maximum values for most of the investigated metrics (Figs. 4 and 5, Table 2). Its location in the high-elevation core of the Alps (Mont Blanc Massif, long- and short-term high uplift rate) was the site of intense Quaternary glaciations (large catchment area above the LGM ELA), which deeply modified the landscape as illustrated by the high geophysical relief of this catchment. Thanks to the highly-resistant granitoid lithology, steep slopes and high reliefs deriving from glacial erosion could be maintained, in turn promoting high millennial to present-day denudation rates in this catchment.

Finally, the supply of sediments by retreating glaciers and active periglacial processes, and the contribution of frequent rockfall events triggered by abundant precipitations (Fig. 5A) and present-day permafrost degradation (Ravanel et al., 2010; Akçar et al., 2012; Deline et al., 2015) participate to the significant sediment yield in the DB01 catchment. Catchment DB01 thus supplies material with highly depleted [10]Be concentrations to the river system, which is in turn capable to significantly dilute the [10]Be signal along the DB course (Fig. 3).

**5.4 Long- and short-term DB denudation rates**

Our [10]Be-derived denudation rates show a general trend of higher millennial denudation rates in the Mont Blanc Massif compared to other DB tributaries (Fig. 2), which is overall in agreement, albeit with different absolute values, with denudation rate estimates on different timescales.

Long-term ($10^6$-$10^7$ yr) exhumation rates estimated from bedrock apatite fission-track data (Malusà et al., 2005) show higher

values (0.4-0.7 km/Myr) in the western sector of the DB catchment (west of Internal Houiller Fault, Fig. 2) than in the east (around 0.2 km/Myr, between the Internal Houiller and the Insubric Faults, Fig. 2). Likewise, results from detrital apatite fission-track (Resentini and Malusà, 2012) indicate that short-term ($10^2$-$10^5$ years) denudation rates are higher (around 0.5 mm/yr) in the Mont Blanc External Massif and its sedimentary cover (west of the Penninic Frontal Thrust) than in the axial belt, east of the Penninic Frontal Thrust (around 0.1 mm/yr; Fig. 2). Similarly to what has been shown by Glotzbach et al.

(2013), the external Alps catchments (west of the Penninic Frontal Thrust; Fig. 2) appear to have equivalent long-term (apatite fission-track derived) and short-term ([10]Be-derived) denudation rates, while internal Alps catchments (east of the Penninic Frontal Thrust; Fig. 2) show higher short-term than long-term denudation rates. This has been tentatively explained by potential differences in driver mechanisms of denudation before and during the Quaternary (Glotzbach et al., 2013). Tectonic forcing dominated Neogene denudation rates, with fast exhuming External Massifs having steeper rivers and higher reliefs and

therefore eroding faster than the slowly-exhuming Internal Alpine Massifs. During the Quaternary, instead, climate fluctuations and associated glaciations modified both the Internal and External Alps morphology and topographic reliefs, also

resulting in increased denudation rates for the Internal Alps. Our [10]Be-derived catchment denudation rates for the DB catchment are therefore not totally reflecting long-term exhumation rates over Myr timescales but most probably highlight Quaternary erosion dynamics.

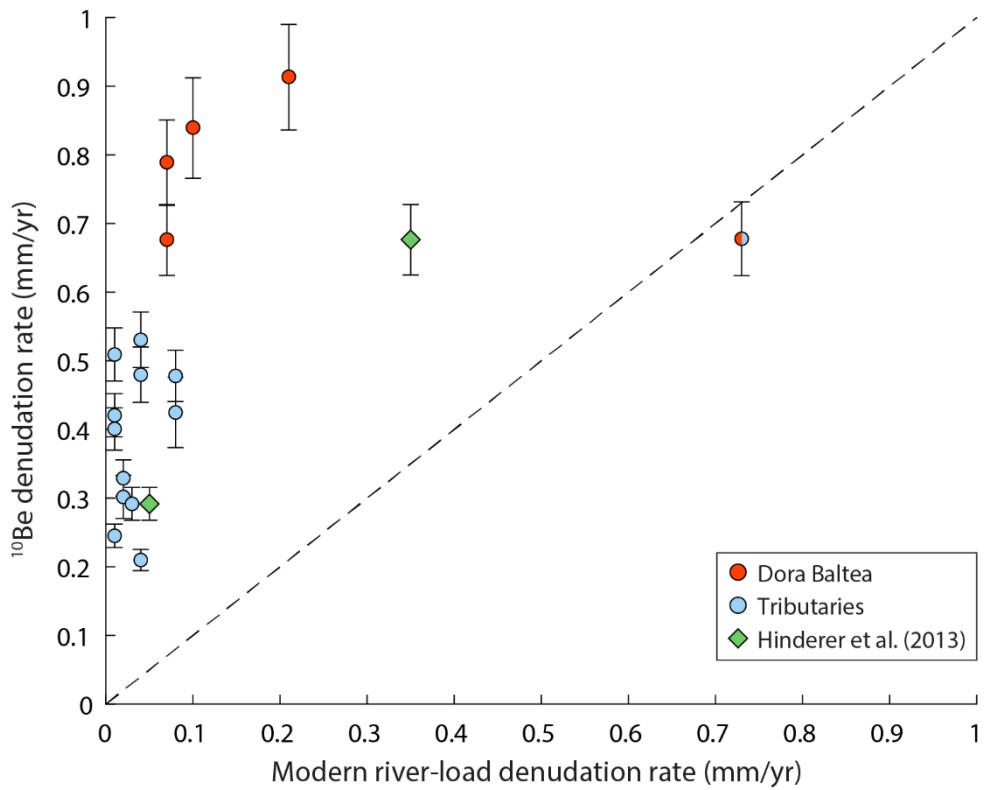

**Figure 8:** Comparison of [10]Be- and modern river yield-derived catchment-wide denudation rates. For all tributary catchments and DB locations, modern denudation rates are derived from river-bedload empirical estimates (Vezzoli et al., 2004; red and light blue dots for DB and tributary catchments, respectively). For catchment DB06 and DB011, also modern denudation rates derived respectively from sediment gauging and sediment trapping are plotted with green diamonds (Hinderer et al., 2013, after Bartolini et al., 1996 and Bartolini and Fontanelli, 2009). Errors are represented only for [10]Be-derived denudation rates, since they are not reported for the modern rates.

Modern denudation rates, obtained from sediment-yield estimates (all DB catchments; Vezzoli et al., 2004; based on Gavrilovic empirical formula, Gavrilovic, 1988) and measurements (sediment gauging for DB06 and sediment trapping for DB11; Hinderer et al., 2003, after Bartolini et al., 1996 and Bartolini and Fontanelli, 2009), display higher values for samples along the DB (0.07-0.73 mm/yr) compared to DB tributaries (0.01-0.08 mm/yr). While such a pattern is consistent with our [10]Be-derived records, millennial denudation rates are 2 to 50 times greater than modern denudation estimates, with the exception of sample DB01 for which modern and [10]Be-derived denudation rates are roughly similar (Fig. 8). Equivalent order of discrepancy between modern sediment-yield and [10]Be-derived denudation rates has been observed by several studies (e.g. Kirchner et al., 2001; Schaller et al., 2001; Wittmann et al., 2007, 2016; Stutenbecker et al., 2018; Pitlick et al., 2021). Among other factors (including the exclusion of non quartz-bearing areas for [10]Be denudation rates, see Table 2 and discussion in section 5.1), this discrepancy was interpreted to point to the separate or combined effects of (1) incorporation of high-magnitude low-frequency erosion events in the [10]Be-derived but not in the sediment-yield denudation rates, (2) contribution of bedload and chemical weathering to [10]Be-derived but not to sediment-yield denudation rates, (3) preferential postglacial erosion of material with low [10]Be concentration, increasing [10]Be-derived denudation rates, through fluvial linear dissection of the landscape and subglacial sediment export, (4) sediment traps (e.g. lakes, dams), changing the flux measured by sediment gauging but less probably the [10]Be concentrations which are averaged over longer timescales. The first and third hypotheses could be the most plausible for our results. Modern denudation rates are potentially not capturing the occurrence of large sporadic erosional events (Kirchner et al., 2001; Schaller et al., 2001), with the exception of catchment DB01 (and therefore

DB02, 06, 10, 12 along the main DB course), where major erosional events have been occurring during the Holocene towards present-day (i.e. rockfall events; Deline et al., 2012, 2015) and therefore potentially included in the $10^1$-$10^2$ yr integration time
of the modern denudation rates. Alternatively, low $^{10}$Be-concentration sediment input in the river system, coming from linear fluvial incision and subglacial sediment export, could explain the mismatch between modern and millennial denudation rates, with $^{10}$Be-derived denudation rates being potentially overestimated (Stutenbecker et al., 2018).

**Conclusions**

Our $^{10}$Be-derived catchment-wide denudation rates obtained in the Dora Baltea (DB) catchment (western Italian Alps) vary
between 0.2 and 0.9 mm/yr and fit within literature values across the European Alps (Delunel et al., 2020). Correlation of output denudation rates with topographic, environmental and geologic metrics excludes any significant control of precipitation and rock uplift on the observed variability in denudation rates within the DB catchment. Our results instead highlight the main influence of catchment bedrock structuration and erodibility (litho-tectonic origin) and resulting topographic metrics on denudation rate variability among the 13 main tributaries. As previously supposed for some other parts of the Alps, our study
shows that the most resistant lithologies (granite and gneiss) support high-elevation and high-relief catchments where glacial and slope processes are more intense and denudation rates are higher than in low-elevation/relief catchments, dominated by "high erodibility" bedrocks (carbonate and terrigeneous rocks).

This litho-tectonic control on catchment denudation is exemplified by the tributary catchment draining the Mont Blanc Massif, which has the highest $^{10}$Be-derived denudation rate from our dataset and appears as an end-member for most of the investigated
metrics. Located in the long-term actively-uplifting core of the European Alps, the Mont Blanc Massif also experienced intense Quaternary glaciations which deeply modified the landscape. Steep slopes and high reliefs could be supported by the highly-resistant granitoid lithology, which in turn have been influencing the millennial to present-day high denudation of the catchment, governed by intense glacial/periglacial processes and recurring rockfall events. In addition, our results also show that the high sediment input from the Mont Blanc catchment dominates the DB sediment flux, contributing to >77% of the
$^{10}$Be signal carried by river sediments along the DB main river, even downstream of multiple tributary junctions. This suggests unequal sediment contribution between tributary fluxes along the DB catchment.

Finally, our $^{10}$Be-derived denudation rates allow for comparison with long-term ($10^6$-$10^7$ yr, from thermochronology) and modern ($10^1$-$10^2$ yr, from sediment budget) denudation rates, showing that, albeit different absolute values, the spatial trend in catchment denudation is overall in agreement over different timescales, with higher millennial denudation rates in the Mont
Blanc Massif compared to the rest of the DB catchment.

**Data availability**

The data used in this study is available upon request, supplementary information are available online at https://doi.org/10.5194/esurf-2021-90.

**Author contributions**

ES, PGV and RD designed the study. ES, PGV and NG performed field investigations and sample collection. ES performed $^{10}$Be cosmogenic sample preparation under supervision of NA, $^{10}$Be production/denudation rate calculation and analyses. MC performed $^{10}$Be measurements. RD performed $^{10}$Be-derived sediment budget calculations. ES wrote the manuscript with input from all co-authors.

**Acknowledgements**

The authors warmly thank F. Magrani for the help during fieldwork. J. Krbanjevic is thanked for support for [10]Be sample preparation. This study was supported by the Swiss National Science Foundation SNSF (Grant PP00P2_170559) and the French ANR-PIA programme (ANR-18- MPGA-0006). We thank three anonymous reviewers and the Associate Editor S. Mudd for insightful comments that helped to improve the manuscript.

**Competing interests**

The authors declare no competing interests.

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
