# Peer review of "Spatio-temporal variability and controlling factors for postglacial denudation rates in the Dora Baltea catchment (western Italian Alps)"

_Earth Surface Dynamics, 2021_

## Author Comment (AC1)

$u^b$

**Elena Serra**
**Institute of Geological Sciences**
**University of Bern**

$b$
**UNIVERSITÄT**
**BERN**

Bern, 11 March 2022

Dear Dr. Simon Mudd,

Dear ESurf Editors and Reviewers,

   We are thankful for the insightful discussions and fruitful suggestions made by the reviewers. Therefore, please find below each reviewers' comment (*blue italic font*) followed by our discussion/reply (black regular font). All orthographic corrections and minor changes in the text were fully considered and are not repeated below.

**Reviewer 1**

**Major Comments:**

*Comment 1.1: Several sentences were long and hard to follow, had wrong punctuation, missing words, or wording that should be revised. I highlight only a few of the mistakes in the line-by-line comments. I think that the authors can improve the readability and language of the manuscript with a thoroughly revised version of the article.*

   We thank the reviewer for this comment and for the wording corrections suggested throughout the main text. All co-authors have carefully read and revised the manuscript to improve its language and readability.

*Comment 1.2: The observation of almost constant $^{10}$Be concentration along the main stem is very interesting and I think it deserves much more attention within the paper. The authors show that the same signal can be seen in sediment gauging data, which indicates that this is not due to differences in quartz fertility. I would suggest to put more emphasis on this observation, expand the mixing model analysis, and provide more detailed discussion on this observation. Currently, the mixing model is in the discussion, but I think it should be in the results.*

*Related reviewer's comment, Line 400-412:* *To me this is the most interesting part of the paper. The mixing model has to go into the results. The high contribution from the Mont Blanc massive is obvious in figure 3, and the reader needs to go through many pages of text to finally see this point addressed. Moreover, if there is a way to get quartz bedrock content estimates for the catchment, the quartz fertility should be included in the mixing model (especially since the higher quartz content of the Mont Blanc massive was already mentioned before).*

*Related reviewer's comment, Line 393-396:* *It is a valid hypothesis. Please, add a calculation to test this hypothesis. It seems like you know all the necessary parameters to set up a mixing model.*

We appreciate the reviewer's interest in the evolution of the river-sand $^{10}$Be concentrations along the DB course. As suggested by the reviewer, we implemented and moved the description and results of the mixing model analyses to section 4.1 of the results (lines 235-249 of the revised text):

*"In order to quantify at first order the relative contribution of the Mont Blanc Massif (represented by the lowest $^{10}$Be concentration of sample DB02; Fig. 3) to the $^{10}$Be signal measured along the DB river, we followed the approach reported in Delunel et al. (2014). River-sediment $^{10}$Be concentrations from tributaries and along the DB river have been first normalised to the SLHL $^{10}$Be production rate (i.e. 4.18±0.26 at $g^{-1}$ $yr^{-1}$), implying that variations in normalised $^{10}$Be concentrations represent the variability in denudation rates only. We then estimated the respective contributions of the Mont Blanc Massif and different tributaries through a mixing model (mass-balance model involving catchment $^{10}$Be concentrations and contributing areas; Delunel et al., 2014) considering (A) the normalised $^{10}$Be concentration for river materials exported from the Mont Blanc catchment, (B) the averaged normalised $^{10}$Be concentration from the upstream tributaries contributing to each sampling points along the main DB river and (C) the normalised $^{10}$Be concentration at the sampling points along the main DB river (DB12, 10, 06). Between our two most upstream DB river samples (DB01 and DB02), we based our model on DB02, which provides a more conservative estimate of the contribution of the Mont Blanc Massif to the $^{10}$Be signal measured along the DB river (i.e. the potential contributions of the tributaries are maximized). By applying this simple model, we find that the Mont Blanc Massif (upstream catchment DB02) contributes to 90, 87 and 77% of the river-sediment $^{10}$Be signal measured respectively at locations DB12, 10 and 06, in line with overall constant $^{10}$Be concentrations measured along the DB course (Fig. 3)."*

Although we tried to estimate the quartz-bearing area for our catchments (Table 2 and Figure S1), we cannot fully evaluate the quartz fertility for bedrock litho-tectonic units which would requires more information. In addition, bedrock quartz fertility would not quantitatively represent the quartz flux in the DB river from each tributary stream. This depends indeed on the mass flux from each stream, which itself depends on many factors (i.e. catchment denudation rate, drainage area, sediment storage, fluvial discharge and dynamics) and not on the quartz bedrock fertility alone, as specified in our revised text (lines 373-376): *"A key factor governing the mixing and flux balance of $^{10}$Be concentrations between river streams is the quartz flux from each stream, which is in turn influenced by (1) catchment denudation rate, (2) drainage area, (3) catchment quartz content (Carretier et al., 2015), (4) sediment storage (e.g. dams, lakes, floodplains reducing mass flux but not changing the $^{10}$Be concentration; Wittmann et al., 2016)."*

We expanded this point following the reviewer's suggestion, and qualitatively acknowledge in the discussion that the ~20% lower quartz content of modern sands (Vezzoli et al., 2004) from tributaries DB09, 11, 16 (all with high $^{10}$Be concentration) could be one of the factors explaining the low $^{10}$Be-signal dominance along the DB course. We also report the estimates of Vezzoli (2004) showing that catchment DB02 contributes to ~62% of the quartz flux of the entire DB catchment, in agreement with the results of our mixing model, suggesting a >77% contribution of catchment DB02 to the DB $^{10}$Be signal (lines 379-383): *"the sediment-provenance studies of Vezzoli et al. (2004) and the sediment-yield estimates of Vezzoli (2004) highlighted that river sands from the Mont Blanc catchment (analogous catchment to DB02) have up to ~20% higher quartz content compared to some other DB tributaries (analogous to DB09, 11, 16; Table S4) and contribute to ~62% of the quartz flux of the entire DB catchment (analogous to DB06; Table S4)."*

Finally, as suggested by the reviewer, we now specified in the text that the differences in quartz fertility are not the main driver of the $^{10}$Be-signal dominance of the Mont Blanc Massif, as the same signal is visible in the sediment gauging data (lines 385-388): *"we propose that the $^{10}$Be-signal dominance of the Mont Blanc Massif along the DB course is mainly driven by its high denudation rather than quartz fertility or area coverage, as illustrated by the similar trend of modern denudation rates derived from sediment gauging (Hinderer et al., 2013; see also discussion in section 5.4)."*

*Comment 1.3: The authors often make links between erodibility of a rock (e.g. its mechanical strength), topography, and the denudation rate, where lower erodibilities are inferred to support steeper topography AND higher denudation rates. It is important to note that this would not be the case in a steady-state landscape. In a steady-state fluvial landscape, differences in erodibility would only be expressed as differences in topography, and denudation would be constant throughout the entire landscape. I understand that the studied landscape was recently glaciated and is likely far from a steady-state topography, however it is important to note that steeper slopes do not necessary equal higher denudation rates.*

*Related reviewer's comment, Line 44-45: Please, be more precise in your formulations. In a steady-state fluvial landscape, erodibility would only govern the steepness of the topography but not denudation. I kind of get what you mean, but here and in other places the formulations should be more precise.*

*Related reviewer's comment, Line 45-47: This statement needs to be revised (see above). Commonly, we assume that denudation rates tend to balance rock uplift rates. If this is true, erodibility only controls topographic steepness and not denudation rates. I understand that this region was heavily glaciated and does not represent a steady-state fluvial topography. However, the way the statements in this paragraph are set up, this is unclear. The paragraph discusses controls on denudation rates, but does not indicate what spatial and temporal scales are being discussed in the second part of the paragraph. I can guess that the authors refer to millennial scale denudation rates on a catchment/landscape scale, but it's better to be precise to avoid misunderstandings.*

*Related reviewer's comment, Line 330-335: Similar to my comment above, I would not call this trend "counterintuitive". Differences in erodibility can be expressed through surface slope, and therefore do not require any impact on denudation rates.*

We thank the reviewer for these insightful comments about the connection between rock erodibility and catchment topography in steady-state landscapes. We have now specified in lines 48-53 of the revised text that the control of bedrock erodibility on millennial denudation rates refers to transient landscapes, as the one investigated in our study: *"In transient landscapes such as recently deglaciated alpine settings, the topographic relief has not reached a steady state equilibrium between rock-uplift and denudation (e.g. Schlunegger and Hinderer, 2003; Delunel et al., 2020) and bedrock lithology may exert a significant control on millennial*

*catchment denudation rates through its structuration and erosional resistance (erodibility, Kühni and Pfiffner, 2001). More resistant lithologies have contrasting potential controls on denudation, (1) either decreasing denudation rates because of rock-mechanical strength (Scharf et al., 2013), (2) or promoting higher denudation rates by sustaining steep topography (Norton et al., 2011)."*

We removed the term *"counterintuitive"* in the discussion section.

*Comment 1.4: The authors argue that the lower erodibility bedrock units allow higher geophysical relief to form, which in turn increases denudation rates. However, the slope distributions between the rock units only show minor differences. The authors point to the higher geophysical relief in external and internal units, but it remains unclear why that parameter should be a better predictor of gravitationally driven physical erosion processes than slope. In the current version of the manuscript, it comes across as if the authors choose to ignore the fact that all rock units exhibit similar slopes. I think a better way of presenting the data would be to either, to try and argue that the differences in slope, while minor, are still close enough to erosional thresholds that they actually matter, OR that the denudation rates mostly depend on elevation.*

We thank the reviewer for this comment. We have now revised the paragraph discussing the role of the three topographic variables (i.e., elevation, slope and geophysical relief) on our observed denudation rate (lines 426-432). In particular, we have extended this section with the following sentences to discuss the clarity of the slope signal relatively to denudation rates distribution: *"While the potential effect of slope alone is here challenging to evaluate as all the tributaries exhibit similar averaged slope values, between ~25-30° (with the exception of DB01 with average slope of ~32°, Fig. 4B), denudation rate exhibits a clear correlation with geophysical relief (Fig. 4C), which is function of both slope and elevation difference (Small and Anderson, 1998; Champagnac et al., 2014). We suggest that slope differences between the investigated catchments, while not significant, are nevertheless still close to threshold values (Fig. 4B; Delunel et al., 2020), which, when combined with elevation differences between catchments, would explain the significant relationship observed between geophysical relief and denudation rates (Fig. 4C)."*

Regarding the relationship between topographic variables and lithologies, after recalculating the distributions of elevation, slope and 5-km geophysical relief of the different litho-tectonic units by including only the quartz-bearing areas (see comment 2.1 of the second reviewer), the

above-mentioned distributions changed slightly. The new interpretation of the data is presented in lines 442-448 of the revised text, where we discuss the potential influence of bedrock resistance on all the three morphometrics, as suggested by the reviewer: *"Our results are in line with this interpretation, with the "low-erodibility" granite of the Mont Blanc External Massif supporting the highest elevation and reliefs and slightly steeper slopes (Fig. 7), where efficient geomorphic processes promote the highest catchment denudation rate (Fig. 6). On the other hand, the "high erodibility" rocks of the Briançonnais cover and of the Piedmont units present low elevation, relief and slope values, and are associated with low denudation rates. High elevation sustained by gneisses and granite of the Internal Massifs (2700 m a.s.l.; Fig. 7A) and slightly steeper slopes supported by gneisses and micaschists of the Austroalpine units and of the Briançonnais basement (30-31°; Fig. 7B) would also drive the moderate denudation rates observed in these three litho-tectonic domains (Fig. 6)."*

*Comment 1.5: All regressions in this study seem to be done with an ordinary least squares regression (OLS). The results from an OLS depend on which variable is defined as dependent and which as independent. I suggest to revise all regressions and use a total least squares (TLS) approach. A TLS is independent of variable definition. This will probably change the r² and p-values of the regressions.*

We appreciate the reviewer's suggestion and are thankful for pointing out the interest of the TLS method, which in our understanding allows to compare two independent variables and takes in account both variable errors. However, we found mitigated results in the literature regarding the advantage of the TLS approach over the OSL when investigating the potential correlation between one dependent and one independent variable with low error (e.g. Kilmer and Rodriguez, 2016; Peprah and Mensah, 2017; Lee et al., 2022). Because we here aim to investigate the distribution of denudation rates (y axis, dependant variable) with respect to different topographic/environmental metrics (x axis, independent variable; e.g. slope, elevation, precipitation, etc.), we therefore maintain the OSL regressions in our calculations.

*Comment 1.6: Figure 7 should be presented in section 4.3, otherwise there is a missing piece in the logical flow. The authors show that denudation rates vary between different rock types, but this variation could just be circumstantial because the distribution of topographic and climatic variables may be heterogeneous among the different rock units. As a reader, I need to know if the*

*faster eroding external units are also steeper, to assess if the external units erode faster because they're steeper or because they have a higher erodibility.*

We have followed the reviewer's suggestion and we now present Figure 7 already in section 4.3 of the results (lines 308-314 of the revised text): *"Finally, we also assessed the potential influence of bedrock lithological properties on catchment morphometry, by evaluating the distribution of elevation, slope and 5-km geophysical relief for the quartz-bearing areas of each individual litho-tectonic unit (Fig. 7). Higher elevations are observed for the External and Internal Massifs (median of 2500-2700 m a.s.l.) compared to the other litho-tectonic units (median of 1900-2200 m a.s.l.; Fig. 7A). The slope distributions appear slightly higher for the External Massif, the Austroalpine units and the Briançonnais basement (median of 31-32°) than for the other units (median of 23-26°; Fig. 7B). The External Massif present the highest geophysical reliefs (median of 1700 m), while all the other litho-tectonic units have similar geophysical relief with medians varying between 1000 and 1300 m (Fig. 7C).*

**Minor comments:**

*Line 29: I suggest to cite the first study applying this technique (Brown et al., 1995)*

We have followed this suggestion by adding the suggested references in line 29 of the revised text: "*Brown et al., 1995; Granger et al., 1996; Bierman and Steig, 1996*".

*Line 44: I think there are much earlier papers than Godard 2014 to make the point that an increase in tectonic uplift increases denudation.*

We now refer also to previous studies addressing coupling between tectonic uplift and increased denudation (lines 47-48): "*e.g. Burbank et al., 1996; Montgomery and Brandon, 2002; Binnie et al., 2007; Godard et al., 2014*".

*Line 98: I assume glaciers covered the entire catchment except for some peaks that were sticking out. If so, this point should be made more clearly, or the LGM ice extent could be on figure 1 (unless it would cover the entire fig. 1 area).*

In lines 106-108 of the revised text, we now describe the general extent of the LGM DB glacial system and we make reference to the work of Serra et al. (in press) where the LGM DB glacier configuration can be found. Lines 106-108: *"The DB catchment was repeatedly glaciated during the Quaternary, with major glaciers covering most of the catchment with the exception*

*of the highest peaks (~3000 km², >1000 m thick; Serra et al., in press) and extending into the Po Plain during the Last Glacial Maximum (LGM, ca. 26-19 ka; Clark et al., 2009)."*

*Line 132: It has been argued that topographic shielding corrections should not be performed in most settings (Di Biase, 2018). Personally, I do not use it anymore and would suggest the same for this study, unless there is a particular reason to stick to the correction. Also, the abbreviation LIA is only defined later in the same paragraph.*

We thank the reviewer for this comment. Although we have seen the recent study of Di Biase (2018) and are aware of the current discussion within the community, we decided to maintain the topographic-shielding correction for consistency with the recent Alpine compilation study of Delunel et al. (2020). This was specified first in the Method (section 3.1.), lines 147-149: *"We acknowledge the recent publication by DiBiase (2018) suggesting no need to correct for topographic shielding when calculating catchment-wide $^{10}Be$ denudation rates. Our $^{10}Be$ production rates were however corrected for topographic shielding to follow a conservative approach similar to the recent Alpine compilation study by Delunel et al. (2020)."*; and second in the Results (section 4.1), in lines 213-215 of the revised text: *"Hereafter, we consider $^{10}Be$ production/denudation rates obtained by applying all corrections (Table 1 and Fig. 2), in order to maintain a conservative approach as in the recent Alpine compilation study (Delunel et al., 2020)."* We also acknowledge that not considering topographic shielding will result in very low differences in production/denudation rate results (lines 150-152): *"As reported in Table 1, mean topographic shielding values obtained within the DB catchment are all very similar (~0.95), implying that neglecting the topographic-shielding correction would result in similar output rates generally within error estimates."*

We now define the abbreviation LIA earlier in the text (lines 144-145).

*Line 141-144: This sentence needs some English revision.*

We rephrased lines 159-161 of the revised text as following: *"In order to estimate shielding correction due to glacier cover, $^{10}Be$ production rates were set to null for areas covered by LIA glaciers (GlaRiskAlp Project, http://www.glariskalp.eu; Fig. S1). This conservative approach assumes sufficient ice thickness for complete cosmic-ray shielding (e.g. Delunel et al., 2010; Wittmann et al., 2007)."*

*Line 144-146: I would appreciate a short sentence explaining how this shielding correction looks like, what percentage area it affects, its magnitude, and how it is calibrated.*

The snow-shielding correction factors ($S_f$) were calculated as function of the average elevation ($Z_m$) of each individual catchment based on the formula $S_f = -3.5e^{-9} \times Z_m^2 - 6.0e^{-5} \times Z_m + 1.0$ from the model of Delunel et al. (2020). As reported in lines 164-165 of the revised text, *"The obtained average snow-shielding correction factors vary between 0.82 and 0.87 and were combined to the topographic-shielding corrections as scaling factors for each sub-catchment."*. This approach is therefore applied to the entire sub-catchment area, since we use the average catchment elevation for calculations. Details about calibration of the snow-shielding model against records of snow-water equivalent thickness for the Swiss and French Alps are provided in Delunel et al. (2020) to which we refer the reader in the present study.

*Line 169: "subjected to more erosion" or simply "erode faster". The paragraph contains several long sentences that would benefit from some language revision.*

Following the reviewer's suggestion, we revised the text and simplified/split sentences in lines 187-188.

*Line 238: For a bivariate regression the r-value is written in lower case.*

We have changed *"$R^2$"* into *"$r^2$"* throughout the text.

*Line 241-2: You could instead simply calculate Cook's distance to evaluate if DB01 significantly affects your regression results.*

Following the reviewer's suggestion, we have calculated the Cook's distance in order to assess if DB01 strongly affects the regression results. This was specified in lines 261-263 of the revised text: *"Cook's distance values were also calculated in order to assess whether DB01 strongly influences the derived correlations as a potential outlier. We selected a threshold value of 3 times for DB01 Cook's distance compared to the data mean Cook's distance."*

*Line 339: As a reader, it's unclear to me what exactly your hypothesis is (based on the sentence before). There are several places were the text is written imprecisely, in the sense that I can guess what the authors mean but it's not formulated explicitly. I suggest to revise the text carefully to avoid such ambiguities.*

Following the reviewer's suggestion, we have revised lines 308-310: *"Finally, we also assessed the potential influence of bedrock lithological properties on catchment morphometry, by evaluating the distribution of elevation, slope and 5-km geophysical relief for the quartz-bearing areas of each individual litho-tectonic unit (Fig. 7)."*

Furthermore, all co-authors have carefully read and revised the manuscript in order to remove potential ambiguities.

*Line 354-364: This paragraph should be revised. The authors suggest that there should be a glacial imprint on denudation rates but have trouble arguing for it based on the regression and eventually leave the reader hanging. Please, be more explicit in your interpretation and feel free to speculate as to why the correlation may not be as good as expected.*

After catchment metrics' recalculation excluding non-quartz-bearing areas (as asked by reviewer 2 and 3, see our detailed answer to main comment 2.1 and 3.2), we find a significant correlation between catchment denudation rates and catchment proportion above LGM ELA, indicating a potential direct control of LGM glaciers' erosional power on our calculated denudation rates. This was specified in lines 457-464 of the revised text: *"Lastly, we consider the potential connection between landscape glacial imprint and catchment denudation rates. Statistically significant correlation between catchment denudation rates and catchment area proportion above LGM ELA (Fig. 5D) suggests an impact of LGM/large Quaternary glaciations on $^{10}$Be-derived denudation rates. Most of the catchments have >50% of their area above the LGM-ELA, indicating that large glaciers persisted during the LGM (and potentially older Quaternary glacial stages), with a significant impact on catchment topography characterized by steep slopes and high reliefs (Fig. 4A-C, Pedersen and Egholm, 2013). We hence tentatively interpret the significant correlation between denudation rates and high elevation / pronounced geophysical relief (Figs. 4B and C) to be indicative of a long-term glacial topographic control on the postglacial erosional response, as suggested by previous studies (Norton et al., 2010a; Glotzbach et al., 2013; Dixon et al., 2016)."*

*Line 365: The authors sometimes start paragraphs with a sentence that ends without being finished. Here the paragraph starts by stating "we propose a hypothesis" but the sentence ends without the proposal. If you want to state the hypothesis in the subsequent sentences you need to add something like "in the following".*

Following the reviewer's suggestion, we have modified lines 470-472 of the revised text: *"By considering the above-mentioned controlling mechanisms for catchment denudation, we propose the following interpretation for the high denudation rate obtained for catchment DB01 compared to other DB tributaries (Figs. 2 and 4-5). Catchment DB01 has maximum values for most of the investigated metrics (Figs. 4 and 5, Table 2)."*

*Line 456: A strong time-scale bias on erosion rates has been shown for glacial environments (Ganti et al., 2016), where decadal scale erosion rates have been shown to be an order of magnitude higher than millennial scale rates due to the stochasticity of erosional processes.*

We thank the reviewer for the suggested reference. However, as highlighted in lines 512-515 of the text, our data rather show higher millennial than decadal denudation rates, in agreement with other previous studies (i.e. Kirchner et al., 2001; Schaller et al., 2001; Wittmann et al., 2007, 2016; Stutenbecker et al., 2018) and contrary to the timescale bias suggested by Ganti et al. (2016). Our observed trend has been also confirmed by a recent publication (Pitlick et al., 2021) that we added to our main discussion.

*Tab 1: I do not understand why the caption of table 1 lists all of the details that are already described in the main text. Please, reduce the text within this caption significantly.*

*Tab 2: Same as for Tab. 1. Do not repeat all the methods in the caption.*

Following the reviewer's suggestions, we shortened the captions of both Table 1 and 2. We moved some of the captions' details in lines 130-143 of the revised text.

*Figure 1: Increase line width of rivers, and size of sampling dots for better visibility. Do not use the colors green and red for the sampling dots, since this is the most common color blindness.*

*Figure 2: Please, increase line width of rivers and symbol size. I suggest to change the color palette, because the current colors are not color-blind friendly.*

We appreciate the reviewer's suggestions. We have increased the line width of rivers and the size of sampling dots in both Figures 1 and 2. We have also changed the colour palette and check colour accessibility with "Proof Colors" tool in Illustrator.

*Figure 3: Maybe enlarge symbol size a little bit.*

We have increased the symbol size.

**Reviewer 2**

**Major comments:**

*Comment 2.1: The only general comment is concerning the way the topographic, environmental, and geological metrics have been calculated. Page 5, line 136, the authors write that they excluded non-quartz-bearing bedrocks for the $^{10}Be$ production rates because they do not contribute quartz to the sedimentary system which is the correct way to approach this calculation. However, on page 5 line 155, when discussing the aforementioned metrics, the authors do not specify if they excluded*

*non-quartz-bearing bedrocks and leave the impression that they indeed included all lithologies in their calculations. If this is the case, I recommend that the authors also exclude non-quartz-bearing lithologies in all their metric calculations (topographic, environmental, and geological); otherwise Mafic and Sedimentary areas can potentially skew the values.*

We thank the reviewer for the constructive review and positive comments, and for pointing out this inconsistency in our approach. We have recalculated all the topographic, environmental and geological metrics excluding the non quartz-bearing areas of each individual catchment. This is now specified in lines 170-171 of the revised text: *"we performed topographic analyses, and extracted environmental and geological variables of the quartz-bearing areas (Fig. S1) for each investigated catchment through an ArcGIS-Matlab routine (Delunel et al., 2020)."*

We highlight here that the values of the different metrics did not change significantly and most of the linear correlations between catchment denudation rates and topographic, environmental and geological metrics maintained the same statistical significance after metrics' recalculation. Statistical significance changed only for the correlations between denudation rate and percentage of area above the LGM ELA. The correlations became significant after metrics recalculation, as specified in lines 265-267 and lines 457-459 of the revised text. Lines 265-267: *"significant linear correlations (i.e. p-value <0.05) both with and without DB01 were obtained between catchment denudation rates and mean elevation (Fig. 4A), 5-km geophysical relief (Fig. 4C), the relative abundance of bare bedrock (Fig. 5B), and the percentage of area above the LGM ELA (Fig. 5D)".* Lines 457-459: *"Statistically significant correlation between catchment denudation rates and catchment area proportion above LGM ELA (Fig. 5D) suggests an impact of LGM/large Quaternary glaciations on $^{10}$Be-derived denudation rates."*

Finally, we have also recalculated the proportion of the different litho-tectonic units within each catchment (Fig. 6) and the distribution of elevation, slope and 5-km geophysical relief of the different litho-tectonic units (Fig. 7) by including only the quartz-bearing areas. This was specified in lines 198-199 and lines 308-310 of the revised text, respectively.

Lines 198-199: *"we estimated the relative proportion of the different litho-tectonic units within the quartz-bearing areas of each catchment".*

Lines 308-310: *"Finally, we also assessed the potential influence of bedrock lithological properties on catchment morphometry, by evaluating the distribution of elevation, slope and 5-*

*km geophysical relief for the quartz-bearing areas of each individual litho-tectonic unit (Fig. 7).”*

Recalculations change the litho-tectonic units' distribution only of catchment DB08, now dominated by rocks from the Austroalpine domain (Piedmont units in the previous version of the manuscript), as shown in the revised version of Figure 6.

We observed a slight change also in the distributions of elevation, slope and 5-km geophysical relief of the different litho-tectonic units calculated by including only the quartz-bearing areas. The new data are presented and interpreted in lines 310-314 and lines 442-448 of the revised text.

Lines 310-314: “*Higher elevations are observed for the External and Internal Massifs (median of 2500-2700 m a.s.l.) compared to the other litho-tectonic units (median of 1900-2200 m a.s.l.; Fig. 7A). The slope distributions appear slightly higher for the External Massif, the Austroalpine units and the Briançonnais basement (median of 31-32°) than for the other units (median of 23-26°; Fig. 7B). The External Massif present the highest geophysical reliefs (median of 1700 m), while all the other litho-tectonic units have similar geophysical relief with medians varying between 1000 and 1300 m (Fig. 7C).”*

Lines 442-448: *“Our results are in line with this interpretation, with the “low-erodibility” granite of the Mont Blanc External Massif supporting the highest elevation and reliefs and slightly steeper slopes (Fig. 7), where efficient geomorphic processes promote the highest catchment denudation rate (Fig. 6). On the other hand, the “high erodibility” rocks of the Briançonnais cover and of the Piedmont units present low elevation, relief and slope values, and are associated with low denudation rates. High elevation sustained by gneisses and granite of the Internal Massifs (2700 m a.s.l.; Fig. 7A) and slightly steeper slopes supported by gneisses and micaschists of the Austroalpine units and of the Briançonnais basement (30-31°; Fig. 7B) would also drive the moderate denudation rates observed in these three litho-tectonic domains (Fig. 6).”*

**Specific comments:**

*Erosion and denudation are both used in this paper. I would advise to either define both terms as they are stricto sensu not the same thing or pick one term and stick with it.*

We followed the reviewer's suggestion and replaced the term *"erosion"* with the term *"denudation"* throughout the text.

*Line 28: a few major citations are missing. It is considerate to cite Brown et al. (1995), Granger et al. (1996), and Bierman and Steig (1996) when the method regarding $^{10}$Be derived denudation rates is brought up.*

We added the suggested citations in lines 29-30 of the revised text.

*Line 33: you provide citations later on for climate and tectonic forcings but not for anthropogenic forcing. Please add some or modify the sentence.*

The initial sentence was modified and we removed the reference to anthropogenic forcing which is not the focus of our present study.

*Line 38: clarify what you mean by "recent timescales".*

This was specified in lines 41-42 of the revised text: *"Over recent timescales ($10^2$-$10^3$ years), climate also exerts a control on denudation rates through precipitation and associated runoff (Moon et al., 2011; Bookhagen and Strecker, 2012)"*

*Line 60: "relatively similar climatic conditions" is at odd with page 3 line 75 "mean annual temperatures range from -10°C (high elevation zones) to 15°C" and page 3 line 76 "precipitations are spatially variable". Please rephrase.*

We thank the reviewer for highlighting this inconsistency. We have now modified the text both in lines 66-68 and in lines 80-86 to highlight that the climatic gradient in temperature and precipitation from valley bottom to high altitude zone is similar within the DB catchment and its tributaries. High temperature and low precipitation occur at the catchments' outlet, while low temperature and high precipitation are found at high elevation in the catchments' source.

Lines 66-68: *"Relatively similar climatic gradients and glacial history but variable bedrock lithology and geodetic uplift within the DB catchment and its tributaries..."*

Lines 80-86: *"Present-day mean annual temperatures range from -10°C in high-elevation zones to 15°C at valley bottoms (Regione Autonoma Valle d'Aosta, 2009). Precipitation varies between the semi-arid conditions prevailing at low elevations in the central part of the DB valley (mean annual precipitation of 400-500 mm/yr) and the wet conditions in the high-elevation internal valleys (Isotta et al., 2014). Higher mean annual precipitation values are observed in the Mont Blanc Massif (around 1800 mm/yr) compared to the north-western and*

*southern sectors of the DB catchment (around 1400 mm/yr for Matterhorn and Monte Rosa area, and around 1150 mm/yr in the Grand Paradiso; Isotta et al., 2014)."*

*Line 65: there is an issue with the legend of figure 1: the elevation color ramp is wrong because of the hillshade. You could also add more information to this figure like the location of quartz-bearing rocks or replace this figure with the figure S1.*

We have removed the elevation colour ramp from the legend of Figure 1. For clarity, we preferred not to add more information on this figure. The reader is referred to Figure S1 when explaining the distinction between quartz-bearing and non-quartz-bearing lithologies. Furthermore, we now also provide the proportion of quartz bearing area compared to the total area for each catchment in the updated version of Table 2.

*Line 184: specify the type of uncertainties.*

This was specified in line 140 of the revised text: *"Denudation-rate uncertainties (one-sigma external) were estimated".*

*Line 149: add in the methods the equation used to derive denudation rates from [10]Be concentrations. A reader should be able to reproduce the data from the raw values ([10]Be concentration and production rates).*

Given that our calculations are based on the GIS-software Basinga, using different spatial rasters, we cannot provide a single equation to derive denudation rates from [10]Be concentrations. The reader is referred to the paper of Charreau et al. (2019) where the equations used in Basinga to derive [10]Be catchment-averaged production/denudation rates are reported.

*Line 212: "integration time" would be a more meaningful heading than "apparent ages". Also please be mindful of the significant figures, the "apparent ages" values are too precise.*

We changed the expression "*apparent age*" into "*integration time*" both in Table 2 and in the revised text (lines 141-142). In Table 2, we also reduced the precision of the integration time estimates.

*Line 215: would it be better to add the contributing quartz surface area of each tributary by varying the size of each circle? It could help drive the point that the DB01 sample is the main contributor to the overall [10]Be signal in the sedimentary system. Please change the symbol of the sample T12 from a circle to something else.*

We changed the symbol of sample T12 into a square. In order not to complicate Figure 3, we preferred not to add the information about the contributing quartz surface area. The reader is referred to Table 2 for such information.

*Line 230: please add in this table the km2 or % of quartz-bearing rock in each drainage area.*

We thank the reviewer for this suggestion. A column was added to Table 2 reporting the percentage of quartz-bearing lithologies in each studied catchment.

*Line 231: I'm curious as to why the authors didn't check for temperatures as a controlling variable. It is mentioned on page 3 line 75 that "mean annual temperatures range from -10°C (high elevation zones) to 15°C" which would be partially compatible with the frost cracking window proposed by Delunel et al., 2010.*

We agree that frost-cracking (as proposed by Delunel et al., 2010) can be important for sediment production and thus catchment denudation rates. In the present study we have already discussed this potential mechanism, especially in section 5.3, but based on the denudation vs. elevation correlation (Fig. 4A) as we think that temperature patterns are reflected in changes in elevations within the catchments. Given the relatively constant latitude of the investigated catchments, temperature can indeed be assumed to be directly dependent on catchment altitude (following a lapse rate of ~0.5°C/100 m which have been stable since the last deglaciation; Ghadiri et al., 2020). This is now specified in lines 194-196 of the revised text: *"Catchment average temperature was not estimated since, at the relatively constant latitude of the investigated catchments, temperature variability directly follows catchment hypsometric distribution and thus relates to catchment elevation which is already investigated in the present study (topographic metric)."*

*Line 264: did you check for a correlation between mean elevation and the lithotectonics units? Looking at Figure 2, it seems like some units are only found at high elevation (External Massifs and Internals Massifs) and you also have a correlation between elevation and denudation rates in Fig. 4A. Could the high denudation rates associated with the External / Internal Massifs be related to an elevation-dependent process (like frostcracking) rather than rock properties?*

We did observe a correlation between mean elevation and litho-tectonic units (Fig. 7A), as specified in lines 310-311 of the revised text: *"Higher elevations are observed for the External and Internal Massifs (median of 2500-2700 m a.s.l.) compared to the other litho-tectonic units*

*(median of 1900-2200 m a.s.l.; Fig. 7A).".* Frost-cracking processes as potential sediment sources for high-denudation rates observed in high catchments are also discussed in section 5.3. In addition, we suggest that the correlation between catchment denudation rates and bedrock litho-tectonic classification might be related to the influence that bedrock lithology has on topography, *"Our results are in line with this interpretation, with the "low-erodibility" granite of the Mont Blanc External Massif supporting the highest elevation and reliefs and slightly steeper slopes (Fig. 7), where efficient geomorphic processes promote the highest catchment denudation rate (Fig. 6). On the other hand, the "high erodibility" rocks of the Briançonnais cover and of the Piedmont units present low elevation, relief and slope values, and are associated with low denudation rates. High elevation sustained by gneisses and granite of the Internal Massifs (2700 m a.s.l.; Fig. 7A) and slightly steeper slopes supported by gneisses and micaschists of the Austroalpine units and of the Briançonnais basement (30-31°; Fig. 7B) would also drive the moderate denudation rates observed in these three litho-tectonic domains (Fig. 6)."* (lines 442-448 of the revised text).

*Line 305: I appreciate the effort the authors made in section 5.1. Could you investigate if the corrections have a significant impact on the correlations calculated with the controlling variables?*

We thank the reviewer for this comment. We can confirm here that the denudation rate corrections did not have a major impact on the correlations calculated with the controlling variables. We obtained similar trends with or without corrected denudation rates and this is specified now in revised section 5.1 (lines 328-329): *"However, we should also note that the investigated correlations between denudation rates and topographic, environmental and geological metrics (Fig. 4-5) remain similar when using non-corrected denudation rates."*

*Line 385: what do you mean by "unequal sediment mixing"? The fact that there are low $^{10}$Be concentrations along the DB river compared to the tributaries does not mean that the mixing is inefficient, especially because the tributaries are relatively small (from 54 km$^2$ to 450 km$^2$ - these values are from table 2, please make sure to add the % of contributing quartz-bearing rocks in each catchment) and thus might not have the capacity to drive the $^{10}$Be concentrations up. One way to strengthen your argument is to check the measured $^{10}$Be concentrations vs the expected $^{10}$Be concentrations along the DB river (see Mariotti et al., 2019 for another example of sediment mixing in the Alps).*

We have changed the expression *"unequal sediment mixing"* into *"unequal sediment contribution"* in lines 368-370 of the revised text: *"The relatively constant low $^{10}$Be concentrations measured for samples DB01, 02, 12, 10, 06 (around 1.2 x10$^4$ at/g, Fig. 3) compared to the tributaries (2.0-4.9 x10$^4$ at/g), and the outcomes of our mixing model indicate unequal sediment contribution (non-balanced sediment budget; Savi et al., 2014) between the main DB stream and its tributaries."*

Moreover, following the reviewer's suggestion, we have now considered the relatively large quartz-bearing drainage area of catchment DB02 (i.e., Mont Blanc catchment) as potential explanation for the constant low $^{10}$Be concentration measured along the DB course. This has been specified in lines 377-385 of the revised text: *"While the Mont Blanc Massif (upstream DB02 catchment) represents only a minor fraction of the total DB catchment area (~18%), its quartz-bearing surface area appear 5-90% larger than for other tributaries (Table 2). (...) Since the occurrence of dams is limited to few catchments (Fig. 1), the high quartz flux and $^{10}$Be-signal dominance of the Mont Blanc Massif along the DB course could derive from (1) its high denudation rate (Fig. 2 and Table 1), (2) its large quartz-bearing drainage area and (3) the high quartz content of the Mont Blanc granitoid (Vezzoli, 2004)."*

We have added a column in Table 2 indicating the percentage of contributing quartz-bearing rocks in each studied catchment.

*Line 461: you should also discuss here the fact that the $^{10}$Be denudation rates are calculated on quartz-bearing rocks only while the modern rates are not lithology dependent.*

We have added a statement in the discussion based on the reviewer's comment. However, as shown in Table S2, corrections for quartz-bearing area have only a minimal effect on catchment-averaged $^{10}$Be production and denudation rates (i.e. only up to 10% difference between uncorrected and corrected results). Therefore, such adopted correction exclusively for $^{10}$Be-derived and not for modern sediment-yield denudation rates does not seem to explain the large discrepancy between millennial and modern denudation rates (i.e. millennial denudation rates 2 to 50 times greater than modern denudation rates).

*Line 487: I don't agree with the assessment that one sub-catchment contributing to 77 % of the $^{10}$Be signal implies poor mixing. The $^{10}$Be signal can be driven by one part of the catchment and still be well mixed if the other sub-catchments export low sediment fluxes. Please rephrase or strengthen your argument.*

Following the reviewer's suggestion, we have rephrased lines 547-548 of the revised text: *"This suggests unequal sediment contribution between tributary fluxes along the DB catchment."*

**Reviewer 3**

**Major comments:**

*Comment 3.1: A problem with applying the basin-wide cosmogenic Be-10 approach to determine erosion rates of glacial/post glacial areas is that you risk violating several of the key assumptions inherent in the method. We might not expect glacial topography to have experienced steady-state erosion for a few multiples of the averaging time and that the concentrations in the surface bedrock/soil cover being eroded are in some approximate equilibrium with the rate of erosion. In addition, these landscapes often contain glacial deposits (moraines, tills, fluvioglacial sediments, etc), which confound the basin-wide approach if they are being introduced significantly into the fluvial system. The Be-10 work I'm familiar with that attempts to constrain erosion in glacial/post-glacial regions has generally tried to understand the amount to which the rates might be biased in such settings by sampling contributing glacial features and different landscape elements within the basins (e.g. Wittmann et. al., 2007; Norton et al., 2010; as cited in the MS), or at least performing some sensitivity analysis of the effects (e.g. Dixon et al., 2016; as cited in the MS). The authors recognize these potential problems, e.g. the end of section 5.1, or 5.4, where they suggest it can explain why the Be-10 derived rates differ from modern sediment export rates. Based on the discord between the low rates of the contributing catchments rates versus the high rates inferred for the trunk stream, I would say potential bias is a fair assessment and so one that needs some consideration in relation to the robustness of the denudation rate results. However, this is not quantitatively addressed and assertions are made about (assumed linear) correlations, or lack of, between the derived denudation rate and topography, lithology, precipitation, etc. This is the main problem I have with the manuscript. I want to be convinced that the Be-10 derived rates are reflecting actual rates of erosion to subsequently accept later interpretations but the manuscript doesn't achieve this. The discussions about sediment mixing and the interesting result about the high concentrations in the tributaries versus the trunk stream are useful and likely valid. However, they are based on results that suggest the application of the technique to derive denudation rates, despite the efforts the authors have gone to in order to constrain appropriate production rates, might be flawed, and this needs tested before making statements about what the denudation rates*

*mean. In my view, major restructuring is required to reduce the manuscripts focus on correlations of the denudation rates with various metrics (that are at best showing weak correlations), and to place more emphasis on the robustness, or not, of the results in this setting, and the implications of their results for sediment mixing. Alternatively, the authors should include convincing support for their interpretation of Be-10 concentrations as valid denudation rates.*

We thank the reviewer for this detailed comment. Concerning input of LGM to Lateglacial glacial sediments to the routing system, we disagree that this will bias our $^{10}$Be-derived denudation rates. Since their deposition and glacier retreat these sedimentary deposits also accumulated $^{10}$Be by exposure to cosmic rays and, if they have been degrading through time, they will contribute to $^{10}$Be input in the sedimentary system. This is different for modern/LIA subglacial sediments, which are incorporated with null $^{10}$Be concentration in the routing system, as taken into account in our study (see section 3.1 for details).

We acknowledge that the topography of our study area is in a transient state (i.e. it has not reach a steady-state equilibrium) since the Dora Baltea catchment was almost entirely glaciated during the LGM and became partly ice free during the Lateglacial. We therefore recognise that our $^{10}$Be-derived denudation rates have been calculated under the assumption of steady-state $^{10}$Be concentration depth profiles without considering the impact of LGM glacial erosion. However, when using the approach of Glotzbach et al. (2014) and Dixon et al. (2016) for our study area, we can estimate a maximum 10-15% overestimate in $^{10}$Be-derived denudation rates, given that (1) most of our study area has been deglaciated already since 12-15 kyr, and (2) the $^{10}$Be-derived denudation rates for our DB catchment and tributaries are relatively high (0.2-0.9 mm/yr), allowing re-equilibrium of $^{10}$Be concentration depth profiles in bedrock already close to steady-state conditions. This outcome is similar to Dixon et al. (2016) results for the Eastern Alps. We thus support the interpretation of our $^{10}$Be-derived denudation rates in lines 353-364 of the revised text: *"Finally, we need to assess the impact of LGM glacial erosion on our $^{10}$Be-derived denudation rates (Glotzbach et al., 2014; Dixon et al., 2016), since our study area has been largely glaciated during the LGM (Serra et al., in press). Deep glacial erosion may have largely to completely zeroed $^{10}$Be concentration on bedrock surfaces, with non steady-state $^{10}$Be concentration depth profiles during postglacial surface exposure leading to apparent overestimate in denudation rates from $^{10}$Be concentrations in river sands (Glotzbach et al., 2014). However, given the deglaciation history of the DB catchment (i.e. largely deglaciated*

*by 14-12 ka; Baroni et al., 2021; Serra et al., in press) and the range of our $^{10}$Be-derived denudation rates (0.2-0.9 mm/yr, Table 1 and Fig. 2), we can estimate an overestimate of our $^{10}$Be-derived denudation rates by 10-15% at maximum, similar to the proposed estimate of Dixon et al. (2016) in the Eastern Alps. We thus are confident in the validity of our $^{10}$Be-derived denudation rates (Table 1) and can exclude any potential strong bias influencing the spatial pattern (Fig. 2) and interpretation with regards to topographic, environmental and geological metrics (Figs. 4-5)".*

Comment 3.2*: It's not entirely clear how lithology/quartz content is being dealt with in regards to the Be-10 approach. Specifically:*

- *L458, how would carbonate dissolution contribute to the Be-10 results?*

Sorry this was a mistake and we were rather relating to "chemical weathering", this is corrected in revised text (lines 518-519): *"(2) contribution of bedload and chemical weathering to $^{10}$Be-derived but not to sediment-yield denudation rates".*

*- Are the areas excluded from contributing to the Be-10 inventory because of lithology also excluded from the topographic (etc) metrics? I don't see this mentioned in the methods.*

We thank the reviewer for this comment. We have recalculated all the topographic, environmental and geological metrics excluding the non-quartz-bearing areas of each individual catchment. This is now specified in lines 170-171 of the revised text: "*we performed topographic analyses, and extracted environmental and geological variables of the quartz-bearing areas (Fig. S1) of each investigated catchment through an ArcGIS-Matlab routine (Delunel et al., 2020).*"

We also highlight here that the values of the different metrics changed only slightly and most of the linear correlations between catchment denudation rates and topographic, environmental and geological metrics maintained the same statistical significance after metrics' recalculation.

- *Are these exclusions not somehow biasing the conclusion that resistant lithologies are a main driver of the rates (assuming the rates are valid, see above)?*

As reported in lines 336-337 of the text, *"correction for quartz-bearing area has only a minimal effect on catchment-averaged $^{10}$Be production and denudation rates, with only up to 10% difference between uncorrected and corrected results thus overlapping within uncertainties (Table S2).*" Therefore, we think that the correlation between catchment denudation rates and bedrock litho-tectonic classification is not biased by the lithological correction. This is visible

from the figure below (modified after Fig. 6 of the main text), where we report both uncorrected and corrected denudation rates of the DB tributary catchments (open and filled circles, respectively; values in Table 1), organized based on catchment lithology. The trend is the same for both corrected and uncorrected denudation rates, with the highest rate for the External Massif, moderate rates for Internal Massif, Austroalpine units and Briançonnais basement, and the lowest rates for the Piedmont units and the Briançonnais cover.

[Figure]

- *Do the exclusions also get taken into account for the sediment mixing/contributing area interpretations, e.g. the suggestion that Mont Blanc regions contributes the most to the downstream sediment yield?*

Yes, the sediment mixing model is based on $^{10}$Be concentrations normalized over "corrected" production rates (including lithological correction), and the area proportion has been also calculated based on the quart-bearing areas of each catchment.

*Comment 3.3: L160 "geophysical relief" needs more explanation. It needs to be more clearly stated how it's derived and how it represents "locally increased erosion".*

The definition and approach for calculating the geophysical relief are fully given in the main text (lines 177-179): "*The geophysical relief (i.e. averaged elevation differences between a surface connecting highest topographic points and the current topography; Small and Anderson, 1998) was calculated in ArcGIS using a 5-km radius sampling window*". We have rephrased the end of the sentence, and refer the reader to Champagnac et al. (2014) for discussion about geophysical relief.

*Comment 3.4: Related to the correlations given in the plots of Figs 4 and 5: why assume linear relationships? We already have a lot of evidence showing, for example, slope and denudation rate are non-linear at such steep slope values?*

We agree that some literature studies have shown non-linear relationships between slope and denudation. However, our aim in the present study is to investigate potential relationships between denudation rates and different topographic, environmental and geological metrics with similar approach, thus choosing linear regression as best-objective approach for the different metrics. Concerning the slope vs. denudation relationship, our results show a correlation (when including DB01, Fig. 4B) with our assumed linear relationship. Going beyond this first-order approach would be outside the scope of our present study. In addition, the non-linear dependence of slope on denudation is discussed in section 5.3.

**Minor comments:**

*On a few occasions I found the text difficult to follow/understand (for example, but not limited to, point (1) on L458) and proof reading of any future versions before submission would be recommended.*

Following the reviewer's suggestion, we have modified the highlighted sentence (lines 517-518 of the revised text): *"(1) incorporation of high-magnitude low-frequency erosion events in the $^{10}$Be-derived but not in the sediment-yield denudation rates."*

All co-authors have carefully proofread and revised the manuscript in order to improve its language and readability.

*Line 31: around the globe covers the Alps*

We have modified lines 31-33 as following: *"Widespread research has used this technique to estimate catchment denudation around the globe (see reviews in Portenga and Bierman, 2011;*

*Willenbring et al., 2013; Codilean et al., 2018) and specifically in mountain belts such as the European Alps (Delunel et al., 2020 and references therein)".*

This was specified in line 41 of the revised text: *"Over recent timescales ($10^2$-$10^3$ years),...".*

We thank the reviewer for pointing this out. We have modified line 80 of the revised text accordingly: *"connected to major >4000-m Alpine peaks".*

*Section 2 (and later section 5.4) over-use parentheses*

We reduced the use of parentheses in both sections.

*Line 109: Is this rate a result, or background info? Better to say - was obtained by Wittmann et al (2016, sample T12). The similar location to the sample measured here can be mentioned in the methods.*

Following the reviewer's suggestions, we have modified both lines 104-105 and lines 121-122 of the revised text.

 Lines 104-105: *"For the entire DB catchment, a $^{10}$Be-derived denudation rate of 0.6 mm/yr was obtained by Wittmann et al. (2016, sample T12)."*

Lines 121-122: *"One sample (DB06) was collected at the same location as sample T12 from Wittmann et al. (2012) to assess for the possible temporal variability of the in-situ $^{10}$Be signal exported by the DB river."*

*Line 124: Is the blank values used from a long-term lab average, or a blank measured at the same time as the samples?*

The blank used to correct $^{10}$Be concentrations was prepared (Be extraction) and measured at the same time as the samples. This is specified with expression *"full process blank"* in lines 128-129.

*Line 159: It's worth acknowledging that slope may be underestimated at the resolution of the DEM. For example, see Zhang and Montgomery (1994) Digital elevation model grid size, landscape representation, and hydrologic simulations. Water Resources Research, 30(4), 1019-1028.*

We thank the reviewer for pointing out this issue. We acknowledge that the use of a 35-m resolution DEM was driven by computational reasons. The use of 'gradient8' Topotoolbox function (Schwanghart and Scherler, 2014) for slope analyses should circumvent the problem

of slope underestimation, since this function returns the steepest downward gradient of the 8-connected neighbouring cells of a DEM. Furthermore, we highlight that our slope results appear comparable to the values obtained in the Alpine compilation of Delunel et al. (2020), using a 90-m resolution DEM (same slope value for our catchment DB06 and catchment T12 from Wittman et al., 2016, calculated in Delunel et al., 2020).

*Line 184: 1.49±0.13 mm/yr is not in the table*

We thank the reviewer for pointing this out. There was a mistake in the uncorrected denudation rate value of catchment DB02 reported in Table 2. We have inserted the value 1.49±0.13 mm yr$^{-1}$ in Table 2.

*Table 1: Give units for the coords (decimal degrees?).*

We have now specified the units of sample coordinates in the heading of the second column of Table 1 (*Location WGS 84 (dd N/ dd E)*) and in the table caption (line 218): *"Sample coordinated are given in decimal degrees (dd)"*.

*Line 403: The Be-10 concentrations were normalized to the basin averaged production rates. Is this not commensurate with simply using the denudation rate (also a normalization of concentration to the production rate)?*

In this section, our aim is to trace sediment sources from the Mont Blanc massif (represented by DB02), which is then progressively diluted by the sediment signature from tributaries input. We preferred to work with $^{10}$Be concentrations rather than with denudation rates. However, variability in $^{10}$Be concentrations is also involving changes in catchment production rates, that we avoided by normalising the $^{10}$Be concentrations by the respective correction factors. This is specified in lines 237-239 of the revised text: *"River-sediment $^{10}$Be concentrations from tributaries and along the DB river have been first normalised to the SLHL $^{10}$Be production rate (i.e. 4.18±0.26 at g$^{-1}$ yr$^{-1}$), implying that variations in normalised $^{10}$Be concentrations represent the variability in denudation rates only."*

*Line 460: How is the landscape dissected by sediment export?*

We have modified lines 519-521 of the revised text for clarity: *"(3) preferential postglacial erosion of material with low $^{10}$Be concentration, increasing $^{10}$Be-derived denudation rates, through fluvial linear dissection of the landscape and subglacial sediment export"*.

**Cited references:**

- Kilmer, J.T. and Rodríguez, R.L. (2017), Ordinary least squares regression is indicated for studies of allometry. J. Evol. Biol., 30: 4-12. https://doi.org/10.1111/jeb.12986

- Ghadiri, E., Affolter, S., Brennwald, M.S., Fleitmann, D., Häuselmann, A.D., Cheng, H., Maden, C., Leuenberger, M. and Kipfer, R., (2020), Estimation of temperature–altitude gradients during the Pleistocene–Holocene transition from Swiss stalagmites. Earth and planetary science letters, 544: 116387. https://doi.org/10.1016/j.epsl.2020.116387

- Lee, J., Lee, W.S., Jung, H. *et al.* Comparison between total least squares and ordinary least squares in obtaining the linear relationship between stable water isotopes. *Geosci. Lett.* **9,** 11 (2022). https://doi.org/10.1186/s40562-022-00219-w

- Peprah, M. S. and I. O. Mensah, (2017). Performance evaluation of the Ordinary Least Square (OLS) and Total Least Square (TLS) in adjusting field data: an empirical study on a DGPS Data, South African Journal of Geomatics, 6(1), 73-89.

- Pitlick, J., Recking, A., Liebault, F., Misset, C., Piton, G., & Vazquez-Tarrio, D. (2021). Sediment production in French Alpine rivers. Water Resources Research, 57, e2021WR030470. https://doi.org/10.1029/2021WR030470

---

## Author Response (AR2)

**Elena Serra**
**Institute of Geological Sciences**
**University of Bern**
**elena.serra@geo.unibe.ch**

Bern, 8 April 2022

Dear Prof. Simon Mudd,

We are thankful that our effort in addressing the reviewers' comments was appreciated and we thank you for the additional suggestions that you raised. Therefore, please find below each of your comments (*blue italic font*) followed by our discussion/reply (black regular font). All orthographic corrections and minor changes in the text were fully considered and are not repeated below.

**General comment:**

*Comment 1: I am happy with the responses to reviewers, and think this paper is nearing readiness for publication. However, I have some remaining concerns that will require one more round of revision. You will see my comments in the pdf, but it boils down to a clearer explanation of why the most upstream DB sample, which is argued to drive continuing high denudation rates downstream, has a lower denudation rate than the downstream samples. There could also be more clarity on the mixing calculations and what they explain. I also had a few minor editorial comments. I look forward to the revised manuscript.*

We thank the editor for the insightful comment and for the text corrections suggested throughout the manuscript, which were all implemented. We have also updated and checked the reference list.

As we now explain in lines 237-242 of the revised text, the lower denudation rate of sample DB01 compared to the downstream DB samples (DB02, 12, 10, 06), despite the similar $^{10}$Be concentrations, is probably due to an overcorrection of $^{10}$Be production/denudation rates of catchment DB01, due to its high elevation and steep topography: *"Because of its generally higher elevation and steeper topography, catchment DB01 shows the maximum production rate corrections for topographic, snow shielding and LIA-glacier cover (Table S2) and consequently yield lower output denudation rate compared to the other sampling locations downstream along*

the main DB river (DB02, 12, 10; Fig. 2), despite similar $^{10}Be$ concentrations (Fig. 3). This suggested overcorrection of the $^{10}Be$ production rate for the DB01 catchment is reflected by the large difference between uncorrected and corrected denudation rates (1.45 and 0.68 mm/yr, respectively, Table 1)."

We have also better clarified how the mixing calculations were conducted in lines 245-258 of the revised text (see reply to comment 5).

**Detailed comments:**

*Comment 2, line 187: Are you just referring to total denudation when you mention physical erosion and chemical weathering? If so, say that. Because it doesn't seem that these two denudation mechanisms will be separated in the paper.*

Following this suggestion, we modified lines 186-188 of the revised text by removing the distinction between physical erosion and chemical weathering: *"Percentage of bare-rock area was estimated from the extent of class 30 ("bare bedrock") of the 100-m resolution CORINE Land Cover Inventory (2018), to consider if catchment areas with null to low soil/vegetation cover have higher denudation rates".*

*Comment 3, line 227: I would add a short note here that based on Figure 2 and the table the erosion rates calculated from these three basins are very similar to the tributaries into which these sub-tributaries drain.*

We followed this suggestion and partly modified the caption of Figure 3 (lines 228-229): *"Samples DB03, 14 and 18 are omitted since they do not directly connect to the main DR river, but note that their $^{10}Be$ concentrations are similar to those of the tributaries into which they drain (DB04 and DB13, respectively; Fig. 2 and Table 1).".*

*Comment 4, line 235: Why is this not captured by DB01, which seems to get most of the Mont Blanc Massif?*

As specified in lines 249-251 of the revised text, in the mixing model we considered sample DB02 rather than DB01 as representative of the contribution from the Mont Blanc Massif, since DB02 has the lowest $^{10}Be$ concentration, allowing therefore to maximize the potential contributions of the tributaries along the DB river: *"We based our model on sample DB02 (lowest $^{10}Be$ concentration) which provides a more conservative estimate of the contribution of*

*the Mont Blanc Massif to the $^{10}$Be signal measured along the DB river (i.e. the potential contributions of the tributaries are maximized)."*

The reason why DB01 has slightly higher $^{10}$Be concentration than DB02 is instead explained in lines 234-236 of the revised text: *"A possible explanation for such a dilution may arise from shielded materials supplied by the incision of a bedrock gorge located between samples DB01 and DB02 sampling sites (and upstream of the tributary junction with catchment DB19)."*.

*Comment 5, line 237:* *Okay, I have a "nesting" erosion rate model (in CAIRN) where you 1) Get the erosion rate of a nested basin 2) then iterate on the erosion rate of the remaining basin in order to 3) arrive at the correct concentration at the outlet of the larger basin. I think this is also what you are doing but it isn't really clear from the text. Can this be clarified?*

We acknowledge models which calculate erosion rates of nested basins (e.g. CAIRN), and refer to this in the revised text. However, in our study we applied a simpler first order approach, based on the mass-balance between the normalized $^{10}$Be concentrations of the Mont Blanc catchment (DB02) and the tributaries. This was better specified in lines 245-258 of the revised text: *"To quantify at first-order the relative contribution of the Mont Blanc Massif to the $^{10}$Be signal measured along the DB river, we followed the approach reported in Delunel et al. (2014), where relative contributions of different sediment sources can be estimated based on their respective $^{10}$Be concentrations. Although other approaches can be used, which consider nested catchments for quantification of sub-catchments denudation rates (Mudd et al., 2016), we adopted a first-order approach in this study based only on relative contributions in $^{10}$Be concentrations between different (sub-)catchments. We based our model on sample DB02 (lowest $^{10}$Be concentration) which provides a more conservative estimate of the contribution of the Mont Blanc Massif to the $^{10}$Be signal measured along the DB river (i.e. the potential contributions of the tributaries are maximized). River-sediment $^{10}$Be concentrations from tributaries and along the DB river have been first normalised to the SLHL $^{10}$Be production rate (i.e. 4.18±0.26 at g$^{-1}$ yr$^{-1}$), implying that variations in normalised $^{10}$Be concentrations represent the variability in denudation rates only. We then estimated the respective river sediment contributions of the Mont Blanc Massif and different tributaries through the mixing model of Delunel et al. (2014; C = xA + yB, x + y = 1) considering (A) the normalised $^{10}$Be concentration for river materials exported from the Mont Blanc catchment (upstream catchment DB02), (B) the averaged normalised $^{10}$Be concentration from the upstream tributaries contributing to each*

*sampling points along the main DB river and (C) the normalised $^{10}$Be concentration at the sampling points along the main DB river (DB12, 10, 06).".*

*Comment 6, line 246: I struggle to understand this. DB01 has a lower basin-wide denudation rate than DB02 according to figure 2. Meaning that the somewhere between DB01 and DB02 there must be something eroding faster than catchment DB01. But the tributary DB19 is eroding more slowly. So there must be a low concentration supply from somewhere else. I do not understand how you get a dilution of $^{10}$Be at DB02 if almost all of it comes from DB01. What am I missing?*

As specified above (see reply to comments 1 and 4), the lower denudation rate of sample DB01 compared to DB02, despite the similar $^{10}$Be concentrations, is probably due to an overcorrection of $^{10}$Be production/denudation rates of catchment DB01, because of its high elevation and steep topography, leading to the maximum corrections for topographic and snow shielding and LIA-glacier cover (lines 237-242). The $^{10}$Be dilution at DB02 is instead related to input of low-$^{10}$Be concentration material between DB01 and DB02, potentially deriving from a gorge located just downstream sample DB01 and therefore not captured in the DB01 $^{10}$Be concentration (lines 234-236).

*Comment 7, line 320: I think it would be useful to include geodetic uplift in this plot. The piedmont has the lowest denudation rates. Presumably it also has the softest rocks. But figure 4 suggests that the geodetic uplift is not the cause of low denudation rates. Seeing this in figure 7 would be useful.*

We have followed this suggestion and added an extra panel (D) in Figure 7, showing the distribution of the mean geodetic uplift rates in the different litho-tectonic domains. We also added few sentences in the revised text describing the absence of correlation between uplift and denudation rates of the different litho-tectonic units.

Lines 326-329: *"We also evaluated the distribution of the mean geodetic uplift rates in the different litho-tectonic domains (Fig. 7D). The highest uplift rates are observed for the Briançonnais basement and cover and for the External Massif (median of 1.0-1.2 mm/yr), while the other litho-tectonic units have uplift rates with median of 0.8-0.9 mm/yr.".*

Lines 468-474: *"No correlation appears instead between the mean geodetic uplift and the denudation rates of the different litho-tectonic units, with the highest uplift rates observed both for the fast eroding External Massif and the slow eroding Briançonnais units (Fig. 7D). This*

*further excludes the role of geodetic uplift in controlling denudation rate variability within the DB catchment (Fig. 4D), while further supporting the dominant influence of bedrock resistance. However, the long-term (late-Miocene) high uplift rates in the Mont Blanc Massif compared to the rest of the DB catchment (Malusà et al., 2005) could be one of the causes of the high denudation rate of the Mont Blanc Massif, by sustaining high-elevations and in turn promoting efficient geomorphic processes."*

*Comment 8, line 385: Okay, why is the denudation rate higher downstream then, after the DB catchment has mixed with more slowly eroding tributaries?*

As specified above (see reply to comments 1 and 6), the higher denudation rate of sample DB02 compared to DB01, despite the similar $^{10}$Be concentrations and the mixing with high-$^{10}$Be concentration sediments from DB19, is probably due to an overcorrection of $^{10}$Be production/denudation rates of catchment DB01 (lines 237-242).

*Comment 9, line 399: The denudation rates vary from ~0.7 to 0.9 mm/yr: are you arguing this is noise and these can be considered a constant rate? State this clearly if that is what you are asserting.*

This was corrected in lines 412-413 of the revised text: *"from the Mont Blanc Massif, only slightly increasing along the DB course (from 1.1 to 1.5 x10⁴ at/g, Fig. 3),".*